# A Novel Resveratrol-Induced Pathway Increases Neuron-Derived Cell Resilience against Oxidative Stress

**DOI:** 10.3390/ijms24065903

**Published:** 2023-03-21

**Authors:** Patrizio Cracco, Emiliano Montalesi, Martina Parente, Manuela Cipolletti, Giovanna Iucci, Chiara Battocchio, Iole Venditti, Marco Fiocchetti, Maria Marino

**Affiliations:** 1Department of Science, University Roma Tre, V.le G. Marconi, 446, 00146 Rome, Italy; patrizio.cracco@uniroma3.it (P.C.); emiliano.montalesi@uniroma3.it (E.M.); martina.parente@uniroma3.it (M.P.); manuela.cipolletti@uniroma3.it (M.C.); giovanna.iucci@uniroma3.it (G.I.); chiara.battocchio@uniroma3.it (C.B.); iole.venditti@uniroma3.it (I.V.); marco.fiocchetti@uniroma3.it (M.F.); 2IRCCS Fondazione Santa Lucia, Via del Fosso di Fiorano, 64, 00179 Rome, Italy

**Keywords:** estrogen receptors, gold nanoparticles, neuronal-derived cells, oxidative stress, resveratrol, signal transduction pathways

## Abstract

A promising therapeutic strategy to delay and/or prevent the onset of neurodegenerative diseases (NDs) could be to restore neuroprotective pathways physiologically triggered by neurons against stress injury. Recently, we identified the accumulation of neuroglobin (NGB) in neuronal cells, induced by the 17β-estradiol (E2)/estrogen receptor β (ERβ) axis, as a protective response that increases mitochondria functionality and prevents the activation of apoptosis, increasing neuron resilience against oxidative stress. Here, we would verify if resveratrol (Res), an ERβ ligand, could reactivate NGB accumulation and its protective effects against oxidative stress in neuronal-derived cells (i.e., SH-SY5Y cells). Our results demonstrate that ERβ/NGB is a novel pathway triggered by low Res concentrations that lead to rapid and persistent NGB accumulation in the cytosol and in mitochondria, where the protein contributes to reducing the apoptotic death induced by hydrogen peroxide (H_2_O_2_). Intriguingly, Res conjugation with gold nanoparticles increases the stilbene efficacy in enhancing neuron resilience against oxidative stress. As a whole, ERβ/NGB axis regulation is a novel mechanism triggered by low concentration of Res to regulate, specifically, the neuronal cell resilience against oxidative stress reducing the triggering of the apoptotic cascade.

## 1. Introduction

In Western societies, higher life expectancy is correlated with an increased incidence of neurodegenerative diseases (NDs) [1]. Therefore, research is aimed at discovering possible treatments to prevent or delay these pathological conditions [2,3,4,5]. Previously, we identified a novel pathway triggered by the sex hormone 17β-estradiol (E2) via the synergic action of both receptor subtypes (i.e., ERα and ERβ) that leads to the accumulation of neuroglobin (NGB) in the mitochondria of different cell types, including cancer and neuronal cells. In these cellular contexts, the activation of the E2/ERs/NGB pathway results in cell protection against reactive oxygen species (ROS)-induced cell death. In particular, in neuron-derived cells (i.e., SK-NE-BE cells), E2 accumulates NGB into the mitochondria, where, upon oxidative stress injury, it binds to cytochrome c (Cyt c), avoiding its release into the cytosol and preventing the subsequent activation of the apoptotic signaling cascade [6,7]. The contemporary activation of both ER subtypes by E2 (in particular ERα activation) renders E2/ERs/NGB pathway ineffective against oxidative stress-induced apoptosis in cellular models of neurodegeneration [8].

Here we hypothesize that ERβ selective ligands, which overcome ERα activation, could increase NGB levels maintaining their protective effects against oxidative stress-induced apoptosis. Two different molecules have been taken into consideration: the diarylpropionitrile (DPN), a synthetic and specific ERβ ligand, and resveratrol (3,5,4′-trans-trihydroxystilbene) (Res), a stilbene family member further identified as a phytoalexin, present in plants such as grape skin, cocoa, various berries and peanuts [9,10,11].

Res is considered a very hopeful molecule for preserving human health [12]. Indeed, Res ameliorates the nervous system’s inflammatory state through the inhibition of pro-inflammatory pathways (inhibition of NF-kB) and the induction of pathways that promote cell survival by acting on SIRT-1 [13]. Moreover, Res has been reported to reduce the production of reactive oxygen species (ROS) and lipid peroxidation [14]. Experiments conducted in neuronal-derived cell lines have also shown that Res treatment prevents rotenone-induced fragmentation of mitochondria by inducing mitochondrial fission and fusion and promoting ATP synthesis [15]. Unfortunately, Res use in clinics encounters several limitations [16]. In particular, the in vivo effects of Res appear to be affected by its low solubility, high biotransformation, and low bioavailability [17,18,19], which leads to the co-presence of a plethora of metabolites that could also exert effects that differ from the precursor and potentially impair its action, as demonstrated for the daidzein metabolites [20]. To further increase the complexity of this picture, Res action pathways are still not fully identified. Among other mechanisms, Res, as well as other polyphenols, is able to modulate estrogen receptors (ERs), acting as an agonist or antagonist of the sex hormone 17β-estradiol (E2), depending on the receptor subtype: i.e., ERα and ERβ [21,22]. Moreover, Res, like E2, increases NGB in neuron-derived cells that mainly express ERβ [22]. Intriguingly, our previous studies on ERα-positive breast cancer cells demonstrated that, unlike E2, Res reduces NGB levels in this cellular context, making these cells more susceptible to chemotherapy-induced apoptosis, thus antagonizing the effects of E2 on ERα, which is the only ER subtype expressed in these cells [22].

At present, the potential of Res to increase NGB levels in the mitochondria of neuronal cells is unknown, as well as the physiological outcomes of this effect. The aim of this work is to assess in neuroblastoma cells SH-SY5Y the effect and action mechanism of this molecule, evaluating the ERβ involvement and the possible maintenance of the neuroprotective Erβ–NGB axis against oxidative stress-induced apoptosis.

## 2. Results

### 2.1. E2-Res Comparison on NGB Levels in SH-SY5Y

First, the expression in SH-SY5Y neuroblastoma cells of estrogen receptor subtypes, ERα and ERβ, has been verified. Figure 1a shows that both receptor subtypes are expressed with a clear predominance of ERβ over ERα.

The cells were treated with different concentrations of E2 (10^−10^ to 10^−7^ M) and Res (10^~7^ to 10^−5^ M) to evaluate NGB levels. Both compounds increase NGB levels at all tested concentrations (Figure 1b,c).

### 2.2. ERs Involvement in Res-Induced NGB Accumulation in SH-SY5Y

Subsequently, to better define the molecular mechanisms underlying the Res effect on the increase in NGB levels, SH-SY5Y cells were treated either with E2 (as positive control), diarylpropionitrile (DPN, a selective ERβ agonist) and with the selective ERα agonist propylpyrazoletriol (PPT). Figure 2a shows that both DPN and PPT stimulation increases NGB levels at lower concentrations than E2, suggesting that both receptors are involved in the E2 action mechanism. In addition, cells pretreatment with (R,R)-5,11-Diethyl-5,6,11,12-tetrahydro-2,8-chrysenediol ((R,R)-THC), the ERβ selective inhibitor, or with endoxifen, the ERα selective inhibitor, reduces NGB levels impairing E2 effect. Altogether, these data argue that both receptor subtype activities synergize in E2-induced modulation of NGB levels in this cell line. On the other hand, as reported in Figure 1, Res still increases NGB levels (Figure 2b). This effect is blocked only in the presence of (R,R)-THC, but not when SH-SY5Y cells are pretreated with the ERα inhibitor endoxifen, suggesting that ERβ activity is sufficient for Res-induced NGB level increase (Figure 2b). Finally, cell costimulation with E2 and Res further confirms that Res and E2 share similar signal transduction pathways (Figure 2c); indeed, this treatment does not further increase NGB levels.

### 2.3. Signaling Pathways Involved in Res Effects

The signaling pathway triggered by the Res/ERβ complex has been investigated. Previous data demonstrated that E2, through ERβ, triggers p38 protein activation, which is crucial for E2-induced NGB accumulation [23]. Therefore, the Res effect on p38 activation has been analyzed in SH-SY5Y cells. Figure 3a shows that there are no variations in p38 total levels, while Res induces p38 phosphorylation that is completely impaired by cell pretreatment with the ERβ inhibitor, (R,R)-THC, or with the p38 inhibitor, SB (Figure 3a). To further confirm the involvement of p38 activation in Res-induced NGB accumulation, SH-SY5Y cells were stimulated with the p38 inhibitor, SB, and NGB levels were determined. Figure 3b shows that the p38 inhibitor significantly reduces the Res effect on NGB levels, further confirming that Res modulatory action on NGB levels requires the activation of the ERβ/p38 pathway in SH-SY5Y cells.

E2 is known to rapidly increase NGB levels by blocking its degradation [23,24]. Therefore, SH-SY5Y cells were treated with MG-132 (MG), the proteasome inhibitor, with chloroquine (Clo), a lysosomal degradation inhibitor, and with cycloheximide (CHX), an inhibitor of gene translation, before Res treatment to verify if this polyphenol acts as an estrogen mimetic also on these pathways. Figure 3c shows that either MG-132 or Clo treatment increases NGB levels with respect to the control, and Res stimulation in the presence of these two inhibitors does not lead to a further accumulation of the protein, suggesting that Res-induced NGB augment depends on the block of NGB degradation. Although CHX treatment enhances NGB levels, its costimulation with Res drastically prevented NGB from the stilbene effects, suggesting that the polyphenol also activates NGB mRNA translation.

### 2.4. Res/ERβ Effect on NGB Localization

As previously reported, the protective effect of NGB in preventing stress-induced apoptosis strictly depends on the increase in its levels in the mitochondria [6]. Therefore, the effect of Res on NGB mitochondrial localization has been investigated. SH-SY5Y cells were treated with E2 (10^−9^ M), Res (10^−7^ M), H_2_O_2_ (5 × 10^−5^ M), and (R,R)-THC (10^−6^ M). After the treatments, the cells were fractionated to separate cytosol and mitochondria. PP2A and COX4 protein levels were used to confirm the purity of the isolated compartments (cytosol and mitochondria, respectively). Res, like E2 and H_2_O_2_, increases NGB levels both in the cytosolic portion (Figure 4a) and in the mitochondria (Figure 4b). Of note, Res pretreatment reduces H_2_O_2_-induced NGB accumulation in both cytosolic and mitochondrial fractions (Figure 4), whereas E2 only reduces the H_2_O_2_ effect in the mitochondria. Intriguingly, cell pretreatment with the ERβ inhibitor (i.e., (R,R)-THC) compromises the Res effect just in the cytosol but not in the mitochondrial compartment, highlighting that Res-induced NGB accumulation is strictly dependent on ERβ in the cytosol while the Res effect in the mitochondria seems to be ERβ independent (Figure 4b).

The NGB level in mitochondria was also confirmed with confocal microscopy, evaluating NGB colocalization (red) with the mitochondrial protein COX4 (green) (Figure 5). Differently from what is reported in Figure 4b, the results indicate that either E2, Res, and H_2_O_2_ increase NGB levels in the mitochondria to the same extent (see bottom panel of Figure 5); however, (R,R)-THC pretreatment does not modify the Res effect on NGB mitochondrial accumulation, confirming the ERβ independence of this Res effect.

### 2.5. Res/ERβ Effect on Cell Resilience to Oxidative Stress

As previously reported, NGB accumulated in the mitochondria of neuronal cells exerts a key function in increasing cell resilience against ROS by counteracting cell death and assuring cell survival [6]. Therefore, we evaluated whether the Res modulation of NGB levels was paralleled with the increase in SH-SY5Y cell survival in the presence of oxidative stress. First, cell viability was analyzed after SH-SY5Y cell treatment with E2 (10^−9^ M), Res (10^−7^ M), H_2_O_2_ (5 × 10^−5^ M), and (R,R)-THC (10^−6^ M) by using propidium iodide (PI) assay. As shown in Figure 6, E2 or Res stimulation increases the DNA amount proportional to the increase in cell number with respect to the vehicle-treated cells. H_2_O_2_, as expected, drastically reduces cell number. Notably, cell pretreatment with E2 or Res provides protection against the reduction of cell number induced by H_2_O_2_ treatment; however, the effects of E2 and Res costimulation are not cumulative, suggesting that a similar pathway is triggered by both the hormone and the polyphenol. Pretreatment with (R,R)-THC does not modify H_2_O_2_ activity or the proliferative effect of Res, while it completely prevents the Res protective effect against H_2_O_2_-induced cell number decrease, supporting the involvement of ERβ in this Res-induced protective pathway.

The propidium iodide assay provides a quick answer regarding the effects of different substances on cell viability but does not allow us to understand if an apoptotic cascade has been activated. Therefore, the level of cleaved PARP-1 (Poly ADP-ribose polymerase), a well-known marker of late apoptosis, was examined. After stimulating SH-SY5Y cells with E2, Res, and H_2_O_2_, we observed that only H_2_O_2_ could induce PARP-1 cleavage (Figure 7a). Moreover, cell pretreatment with E2 or Res before H_2_O_2_ stimulation decreases ROS-induced PARP-1 cleavage. Notably, Res takes back the H_2_O_2_-induced PARP-1 cleavage at the control levels, while E2 just reduces by 2 times PARP-1 cleavage, sustaining its neuroprotective role against oxidative stress-induced apoptosis. Furthermore, cells pretreatment with (R,R)-THC before H_2_O_2_ stimulation strongly reduces the Res antiapoptotic effect (Figure 6b).

### 2.6. Effect of Res Conjugated with Gold Nanoparticles on NGB and on Cell Resilience to Oxidative Stress

As already mentioned in the introduction, one of the main problems related to the use of polyphenols in general, and of Res in particular, lies in their poor bioavailability, mainly due to the extensive metabolism those compounds are exposed to in the human body. Nowadays, an important component of drug research efforts is aimed at improving the pharmacokinetic properties and/or enhancing the bio-efficacy of compounds, such as polyphenols. Recently, Res conjugated with gold nanoparticles has been efficiently synthesized and characterized in terms of toxicity and NGB modulation in breast cancer cells expressing only the subtype α of ER [25,26]. According to these previous results, we tested the effect of different concentrations of Res in conjugation with gold nanoparticles (NP-R) in SH-SY5Y. Results indicate that NP-R significantly increases NGB level already at an NP concentration of 1 µg/mL (corresponding to 10^−8^ M of loaded Res) and 3 and 9 µg/mL NP concentration (corresponding to a load of Res of 3 × 10^−8^ M and 10^−7^ M, respectively) (Figure 8a), while unconjugated nanoparticles (NPs) at the same concentrations do not affect NGB levels. In addition, already at the concentration of 10^−8^ M, Res conjugated with nanoparticles shows the same effect of the higher concentration (10^~7^ M) of unconjugated Res on cell vitality, whereas NPs do not significantly modify the number of cells, indicating that the nanospheres do not possess any toxic or proliferative effects in these cells (Figure 8b).

Finally, the ability of Res conjugated with nanospheres to protect cells against oxidative stress-induced apoptosis was evaluated. The data (Figure 8c) show that, similarly to E2 and unconjugated Res, cell pretreatment with NP-R reduces the H_2_O_2_-induced PARP-1 cleavage already at 10^−8^ M of Res.

## 3. Discussion

Over the past two decades, many studies have been conducted regarding the beneficial power of Res on health [27,28,29,30,31,32,33,34,35]. Among others, Res appears to have a role associated with slowing down or preventing cognitive impairment. In this context, the anti-inflammatory and antioxidant effects of Res lead to the hypothesis that this polyphenol may be a useful treatment for neurological disorders such as Alzheimer’s disease (AD) and stroke that occur through inflammatory damage and oxidative stress to the central nervous system [36]. Furthermore, it has also been shown that Res can have a protective effect on the blood-brain barrier in pathological conditions such as multiple sclerosis [37,38]. Since there are many protective effects ascribed to Res, research has focused on identifying the mechanisms of action by which this stilbene acts. Among those, the direct non-enzymatic antioxidant ability of Res has been hailed as responsible for the neuroprotective effects of this stilbene. Moreover, Res has been reported to increase the activity of SIRT1, acting as a protective molecule against mitochondrial dysfunctions [11,39,40,41]. Finally, the interaction of Res with estrogen receptors has also been demonstrated. Indeed, it was seen that mouse ovarian cells transfected with ERα or ERβ and treated with Res (10^−4^ M) showed reduced cell proliferation compared to non-transfected ones, indicating that the effect was dependent on the expression of the ERs [42]. Furthermore, Res at low concentrations (10^~7^, 10^−6^, and 10^~5^ M) reduced NGB basal levels in ERα-positive breast cancer cells, specifically by antagonizing E2 effects [22].

One of the major criticisms made by the scientific community to the concept of nutraceuticals (i.e., the use of substances of natural origin as possible drugs) lies in the fact that despite the encouraged in vitro and in vivo outcomes when translated to clinical trials, Res did not show the expected significant therapeutic effects [16]. Res is subjected to intense metabolism by the enzymes present in the intestinal microbiota, on the intestinal epithelium, and in the liver, which considerably reduces its bioavailability and increases the concentration of its glucuronidated or sulfated metabolites, whose effects are currently unknown [43,44,45]. Thus, the poor reproducibility of the beneficial effects of Res is mainly associated with the effective concentrations of this compound that have been tested in the literature. It has been estimated that 25 mg of oral dose produces < 5 ng/mL, about 21 nM, of un-metabolized Res in human plasma ([11] and literature cited therein), whereas the majority of studies have been conducted by using a Res concentration ranging from 10^−5^ to 10^−4^ M. In the last decade, to improve the poor bioavailability of this stilbene, various methodological approaches have been developed, such as nanoencapsulation in lipid nanocarriers or the synthesis of nanoparticles [25,26,46,47,48].

The identification of NGB as a neuroprotective protein and the discovery of its positive modulation by E2 in neuron-derived cells have paved the way for the study of new possible action mechanisms for Res. Indeed, it is known that the E2/ERβ complex leads to rapid activation of the p38 kinase with subsequent up-regulation of NGB in the mitochondrion, where the protein prevents the activation of the apoptotic cascade and acts as a stress sensor [23]. This work aims to evaluate whether Res, at concentrations compatible with those present in the plasma following its ingestion [11], is able to activate the ERβ/NGB pathway, inducing the accumulation of NGB and increasing the resistance of neuronal-derived cells to oxidative stress.

From the reported results, it emerges that, already at the concentration of 10^−7^ M, Res is more effective than E2, at the same concentration, in increasing NGB levels and that this mechanism is dependent on ERβ. In fact, by pretreating the cells with (R,R)-THC, a selective inhibitor of ERβ, and endoxifen, an inhibitor of ERα, the effect of Res is blocked only in the presence of (R,R)-THC. Interestingly, in ERα-positive breast cancer cells (MCF-7 and T47D), Res acts as an ERα antagonist by reducing the ability of this receptor subtype to activate AKT kinase phosphorylation and reducing NGB levels [49]. In cells of neuronal derivation used in this work, Res acts as an agonist of E2 on ERβ, the most expressed estrogen receptor subtype in neurons, leading to the accumulation of NGB without promoting the activation of ERα. The reported data indicate that DPN, the synthetic selective ERβ agonist, increases NGB levels to a lesser extent than E2. It would have been expected that the Res, by activating only one receptor subtype, should show effects similar to those observed after the stimulation with DPN rather than having significantly greater effects on the accumulation of NGB. These discrepancies can be reconciled by the evidence that both ERα and ERβ activate discordant signaling pathways in all cell lines, including neurons. In fact, the binding of PPT or E2 to ERα triggers the persistent activation of the PI3K/AKT and ERK/MAPK signaling pathways, which inhibit the activation of the p38 pathway that is activated by the binding of E2 or DPN to ERβ [6,23,50,51,52]. While PPT and DPN activate one or the other pathway, E2, by binding both receptor subtypes, leads to a balance between the divergent signal pathways activated by the two receptor subtypes. Finally, Res inhibits the PI3K/AKT and ERK/MAPK pathways activated by ERα [49]. allowing the full activation of the pathway activated by ERβ. In fact, the reported results show that Res activates p38 kinase by increasing the levels of its phosphorylation in an ERβ-dependent manner; in fact, the effect of Res is blocked when cells are pretreated with (R,R)-THC and SB (p38 inhibitor). In addition, the induction of NGB levels by Res also involves the activation of p38. In fact, in the presence of an SB inhibitor, Res’s ability to increase NGB levels is significantly reduced. Finally, the cell co-treatment with E2 and Res does not lead to a further increase in NGB as the activation of ERα induced by E2, which has a major affinity for its receptors than Res, reduces the effects of ERβ-dependent signaling pathways. It is reported in the literature that E2 is able to increase NGB levels mainly by blocking its degradation and partly by increasing its synthesis [23,24]. The data reported in this work show that, likewise, E2 also Res increases NGB levels by blocking its degradation and increasing *Ngb* gene transduction; in fact, in the presence of either proteasome inhibitor (i.e., MG-132) or the inhibitor of lysosomal degradation (i.e., Clo), Res does not lead to a further accumulation or degradation of NGB to the lysosomal degradation is still working and vice versa. Surprisingly, the protein synthesis inhibitor (i.e., CHX) induces an increase in NGB levels, whereas the CHX costimulation with Res does not increase NGB levels. The *Ngb* promoter lacks the TATA box; however, many conserved putative transcription factor binding sites have been identified within the human Ngb promoter, including two GC-boxes and two neuron-restrictive silencer element (NRSE) sites that are bound by the neuron-restrictive silencer factor (NRSF), a 210 kDa glycoprotein containing nine zinc finger domains, a silencer of neuron-specific gene expression in undifferentiated neuronal progenitor cells [50]. In addition, the *Ngb* promoter region is characterized by the presence of binding sites for several transcription factors such as the specificity protein (Sp) family members Sp1 and Sp3, the cAMP response element binding (CREB) protein, the early growth response protein 1, and members of the Nuclear Factor κ-light chain enhancer of activated B cells (NF-κB) family (i.e., p50, p65, cRel) [50]. It could be possible that 24 h of cycloheximide treatment prevents the translation of the silencer protein, enhancing NGB levels. In addition, it has been reported that cycloheximide alters protein degradation through activation of protein kinase B (PKB/AKT) [51], further enhancing NGB levels. On the other hand, resveratrol triggers the p38 activation, a well-known inducer of several transcription factors including CREB activation [52,53,54] that could overcome NRSF inhibition allowing *Ngb* transcription. Although Res stimulation prevents NGB degradation and enhances *Ngb* transcription, the contemporary treatment with cycloheximide reduces the NGB translation and the NGB protein overexpression could not occur. Altogether these data indicate that, via ERβ signaling, nM concentrations of Res (10^−7^ M) activate, like E2, the pathways that culminate with the accumulation of a neuroprotective protein, i.e., NGB.

Since the protective effect of NGB in preventing stress-induced apoptosis in neuron-derived cells does not depend only on the increase in its levels but also on its intracellular localization, the subcellular localization of this protein was analyzed. The results show that Res increases NGB levels in the cytosol via an ERβ-dependent mechanism. This event is in agreement with what has been shown in the literature, that NGB accumulated in the cytosol by different inducers (e.g., E2 or H_2_O_2_) can be released outside cancer cells [55] or astrocytes [55,56], exerting protective effects on the survival of neighboring cells with autocrine and paracrine mechanisms. In the mitochondria, Res signals are able to carry the globin inside the mitochondria, even if to a lesser extent than E2, and in this latter compartment (R,R)-THC does not seem to influence the activity of Res. Although NGB does not have any mitochondrial localization domains in its sequence, E2 has been shown to induce translocation of NGB into this organelle by increasing huntingtin protein (Htt) intracellular levels through the activation of ERα. Since Res prevents the rapid activities of this receptor subtype, the translocation of NGB to the mitochondrion could rely on other transport proteins whose recruitment could depend on one of the different signaling pathways activated by Res. Once accumulated into the cytosol and in the mitochondrion, NGB can bind the oxidized cytochrome c, released in the cytosol following oxidative stress, avoiding its bond with APAF-1, or it can directly block the cytochrome c inside mitochondria, inhibiting both cases, the activation of caspase 3 and the subsequent apoptotic cascade [6]. The results obtained from the propidium iodide fluorescence assay show the involvement of ERβ in the protective pathway activated by Res against H_2_O_2_-induced cell death. This is further supported by the data obtained from the study of PARP-1 protein as a marker of late apoptosis. As shown, when cells are pretreated with (R,R)-THC before stimulation with Res and H_2_O_2_, the protective role of Res is lost due to the (R,R)-THC-induced blockade of ERβ. Thus, although Res induces NGB accumulation into the mitochondria through an ERβ-independent pathway, the stilbene protection against oxidative stress-induced apoptosis requires the activation of ERβ signaling. Altogether these data highlight a novel protective pathway triggered by Res against oxidative stress injury that involves both ERβ signaling and NGB accumulation in neuron-derived cells and is not linked just to the antioxidant ability of Res chemical structure. Of note, the different modulation of NGB levels triggered by Res in the presence of the two different ER subtypes ([22,26] and present data) open a novel avenue in the field of pharmacological treatment of degenerative diseases.

Although in this work, for the first time, a very low concentration of Res (10^−7^ M) has been used, we adopted a specific strategy to potentially preserve Res from the extensive metabolism to which polyphenols are exposed and to improve its biological activity [57,58,59] by using Res conjugation with highly hydrophilic gold nanospheres as nanocarriers. The reported data demonstrate that neither naked nor Res-conjugated nanoparticles are toxic in SH-SY5Y cells. Moreover, Res-conjugated nanospheres increase NGB levels and maintain the protective effect against apoptosis induced by oxidative stress, preventing PARP-1 cleavage already at 10^−8^ M, then surpassing the effect of unconjugated Res (10^−7^ M), indicating that the conjugation of Res with gold nanospheres not only may enhance Res bioavailability but also strengthens its bioactivity, providing encouraging outcomes for the translation of this different administration routes into clinical application for neuronal protection.

Altogether, these data indicate that the Res effect does not depend on generic antioxidant properties ascribed to the structure of this molecule and, more generally, to all polyphenols of plant origin, which require high intracellular levels, but on the specific activation of ERs signaling pathways, which can be obtained at much lower concentrations, compatible with its bioavailability. The obtained results allow us to consider Res, at concentrations compatible with those present in the plasma following a diet rich in this stilbene, as a substance capable of mimicking the neuroprotective actions of estrogens without the secondary effects of the hormone. In conclusion, Res appears to be a promising molecule capable of maintaining high levels of NGB in neuronal-derived cells, leading to protection against oxidative stress and finding a therapeutic application for this molecule against pathological conditions such as neurodegenerative diseases.

## 4. Materials and Methods

### 4.1. Reagents

The Bradford protein assay and the chemiluminescence reagent for Western Blot Clarity Western ECL Substrate were obtained from Bio-Rad Laboratories (Hercules, CA, USA). The anti-phospho-p38 (Thr180/Tyr182) (cat. n° 4511), anti-p38 (cat. n° 8690), and anti-poly[ADP-ribose] polymerase 1 (PARP-1) (cat. n° 9542) antibodies were purchased from Cell Signalling Technology Inc. (Beverly, MA, USA). The anti-α-tubulin (cat. n° T6074) and anti-COX-IV (cat. n° ZRB1593) antibodies were purchased from Sigma-Aldrich (St. Louis, MO, USA). Anti-PP2A (cat. n° sc-56954), anti-ERα (cat. n° sc-8005), and anti-ERβ (cat. n° sc-373853) antibodies were obtained from Santa Cruz Biotechnology (Santa Cruz, CA, USA). The anti-NGB antibody (cat. n° MABN369) was purchased from Merck Millipore (Darmstadt, D). The ERα inhibitor endoxifen and the ERβ inhibitor (R,R)-5,11-Diethyl-5,6,11,12-tetrahydro-2,8-chrysenediol ((R,R)-THC) were purchased from Tocris (Ballwin, MO, USA). All the other products were from Sigma-Aldrich. Analytical and reagent-grade products were used without further purification.

### 4.2. Cell Culture

Human neuroblastoma cells SH-SY5Y (American Type Culture Collection, LGC Standards S.r.l., Milan, Italy) were grown at 37 °C in air containing 5% CO_2_ in either modified, phenol red-free, Dulbecco’s Modified Eagle’s Medium (DMEM) medium. Ten percent (*vol*/*vol*) of charcoal-stripped fetal calf serum, L-glutamine (2 mM), non-essential amino acid (2×), sodium pyruvate (1 mM), HEPES (10 mM), and penicillin (100 U/mL) were added to the media before use. Cells were used in passages 4–8, as previously described [7]. The cell line authentication was periodically performed by amplification of multiple short tandem repeat loci by BMR genomics S.r.l (Padova, Italy). Cells were treated for 24 h with either vehicle (DMSO or phosphate-buffered saline [PBS], 1:1000; vol/vol) or E2 (10^−9^ M) or Res (10^−7^, 10^−6^, and 10^−5^ M) or H_2_O_2_ (5 × 10^−5^ M). When indicated, endoxifen (10^−6^ M) or (R,R)-THC (10^−6^ M), as indicated by the manufacturer, were added 1 h before Res (10^−7^ M) or E2 administration, Res (10^−7^ M) or E2 were added 4 h before H_2_O_2_ addition for 24 h, while the p38 inhibitor SB203580 (5 × 10^−6^ M) was added 1 h before Res (10^−7^ M). For blocking NGB degradation, the cells were treated for 4 h with the proteasomal inhibitor MG-132 (10^−6^ M), chloroquine (10^−5^ M), and the protein synthesis inhibitor cycloheximide (10^−5^ M), and when in costimulation, Res was administrated at the same time of the inhibitors.

### 4.3. Western Blot Assay

After the treatments, cells were lysed, and proteins were solubilized in the YY buffer (50 mM HEPES at pH 7.5, 10% glycerol, 150 mM NaCl, 1% Triton X-100, 1 mM EDTA, and 1 mM EGTA) containing 0.70% (*wt*/*vol*) sodium dodecyl sulfate (SDS). Total proteins were quantified using the Bradford protein assay. Solubilized proteins (40 μg) were resolved by 7%, 10%, or 13% SDS-polyacrylamide gel electrophoresis at 100 V for 1 h at 24.0 °C and then transferred to nitrocellulose with the Trans-Blot Turbo Transfer System (Bio-Rad) for 7 min. The nitrocellulose was treated with 5% (*wt*/*vol*) bovine serum albumin or non-fat dry milk in 138 mM NaCl, 25 mM Tris, pH 8.0, at 24.0 °C for 1 h and then probed overnight at 4.0 °C with either anti-NGB (final dilution, 1:1000), anti-phospho-p38 (final dilution, 1:1000), anti-PP2A (final dilution, 1:1000), anti-ERα (1:1000), anti-ERβ (1:1000), or anti-PARP-1 (final dilution, 1:1000) antibodies. The nitrocellulose was stripped by the Restore Western Blot Stripping Buffer (Pierce Chemical, Rockford, IL, USA) for 10 min at room temperature and then probed with either anti-p38 (final dilution, 1:1000), anti-COXIV (final dilution 1:1000), or anti-α-tubulin (final dilution, 1:40,000) antibodies to normalize the protein loaded. The antibody reactivity was detected with ECL chemiluminescence Western blotting detection reagent using a ChemiDoc XRS+ Imaging System (Bio-Rad Laboratories, Hercules, CA, USA). The densitometric analyses were performed by the ImageJ software for Microsoft Windows (National Institute of Health, Bethesda, MD, USA).

### 4.4. Mitochondria/Cytosol Fractionation

Cell fractionation was performed using the Mitochondria/Cytosol Fractionation Kit (ABCAM, Cambridge, U.K.). After seeding 10 million cells and treating them as reported above, the medium was removed, and the cells were collected in 15 mL tubes and centrifuged at 600× *g* for 5 min at 4 °C. Subsequently, the supernatant was removed, and the pellet was resuspended in 5 mL of cold PBS. It was centrifuged at 600× *g* for 5 min at 4 °C. The supernatant was removed, and the pellet was resuspended with 1 mL of Cytosol Extraction Buffer Mix 1×. It was incubated on ice for 10 min, and then the cells were homogenized with the Potter. The homogenate was transferred to 1.5 mL tubes and centrifuged at 700× *g* for 10 min at 4 °C. The supernatant was collected in other 1.5 mL tubes and centrifuged at 10,000× *g* for 30 min at 4 °C. The supernatant, i.e., the cytosolic fraction, was taken and placed in other tubes, while the pellet, i.e., the mitochondrial fraction, was resuspended with YY lysis buffer, and the samples were prepared for the electrophoretic run. In the case of cell fractionation, COX4, a mitochondrial protein, and PP2A, a protein localized in the cytosol, were used as control proteins. The detection of these proteins in the cytosolic and mitochondrial fractions provides control of the purity of the obtained fractions.

### 4.5. NGB Mitochondrial Localization with Confocal Microscopy

After having seeded the cells on slides and stimulated with the treatments, the cells were fixed with 4% paraformaldehyde for 10 min and subsequently permeabilized with 0.1% Triton for 5 min. Then the slides with the cells were incubated with 2% bovine serum albumin (BSA) solution in PBS for 30 min. Thus, the slides were incubated overnight at 4°C in the dark with anti-NGB, and anti-COX-IV prepared 1:200 in 0.2% BSA in PBS. The next day, the slides were incubated with Alexa Fluor 488 (anti-rabbit) and 578 (anti-mouse) secondary fluorescent antibodies (Invitrogen, Carlsbad, CA, USA) (red for NGB and green for COX4) prepared 1:400 in 0.2% BSA in PBS for 30 min in the dark at RT. Finally, the slides with the cells were mounted on specimen slides, and the images were captured under the confocal microscope. Images were analyzed with ImageJ software (NIH, Bethesda, MD, USA) and JaCoP plugin (Just another Colocalization Plugin) to determine Pearson’s correlation coefficient.

### 4.6. Cellular DNA Content—Propidium Iodide (PI) Assay

SH-SY5Y cells were grown up to 80% confluence in a 96-well plate and treated with the selected compounds. The cells were fixed and permeabilized with cold EtOH 70% for 15 min at −20 °C. EtOH solution was removed, and the cells were incubated with PI buffer for 30 min in the dark. The solution was removed, and the cells were rinsed with PBS solution. The fluorescence was revealed (excitation wavelength: 537 nm; emission wavelength: 621 nm) with TECAN Spark 20 M multimode microplate reader (Life Science, Italy).

### 4.7. Synthesis and Purification of Gold NP and NP-R

The gold NPs stabilized with citrate and L-cystein (L-cys) were prepared and characterized in analogy to literature reports [25,26,57]. Briefly: 25 mL of L-cys solution (0.002 M), 10 mL of citrate solution (0.01 M), and 2.5 mL of tetrachloroauric acid solutions (0.05 M) were mixed sequentially in a 100 mL flask, provided with a magnetic stir. After degassing with Argon for 10 min, 4 mL of sodium borohydride solution (0.00008 M) was added, and the reaction continued for 2 h at room temperature. Then, the solid brown product was purified by centrifugation (13,000 rpm, 10 min, 4 times with deionized water).

NP-R synthesis was carried out following the same procedures, but including RSV water solution (1 mL 0.02 M) in the reagent mixture, before reduction.

### 4.8. Statistical Analysis

The statistical analysis was performed by Student’s *t*-test to compare two sets of data using the INSTAT software system for Windows. In all cases, *p* < 0.05 was considered significant.

## Figures and Tables

**Figure 1 ijms-24-05903-f001:**
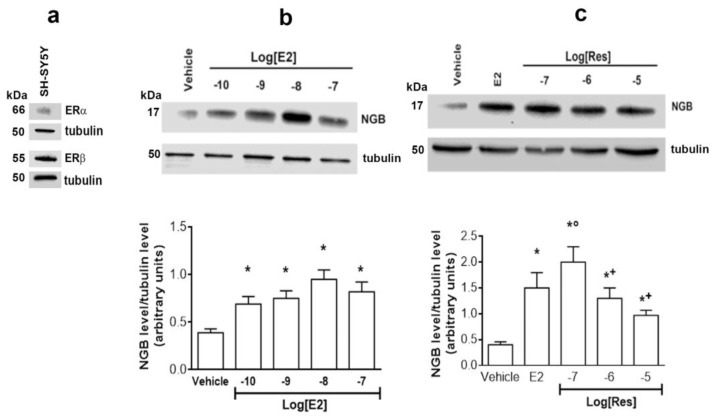
Modulation of NGB levels by E2 and Res. (**a**) Western blot analysis of the levels of the two receptor subtypes ERα and ERβ in SH-SY5Y cells. (**b**,**c**) Western blot analysis of neuroglobin (NGB) levels in SH-SY5Y cells treated for 24 h with different concentrations of E2 (10^−10^–10^−7^ M) (panel b) or Res (10^−7^–10^−5^ M) (panel c) or with vehicle (DMSO 1:1000, *v*:*v*). Tubulin protein was used to normalize blots. The upper panels show a Western blot type, and the lower panels report the densitometric analysis of at least three different experiments. Data are the mean ± SD. *p* < 0.01 was calculated with Student’s *t*-test with respect to vehicle (*) or with respect to E2 (°) or with respect to Res 10^~7^ M (+).

**Figure 2 ijms-24-05903-f002:**
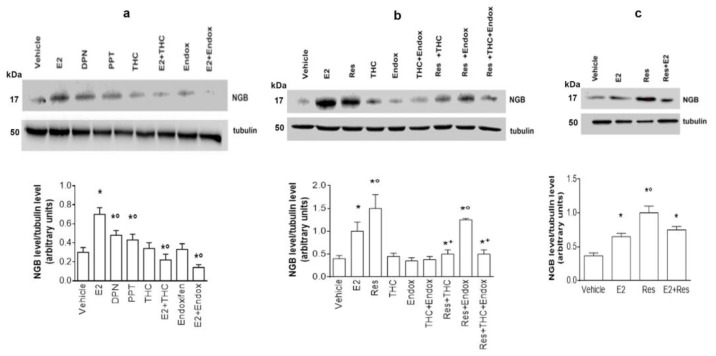
Action mechanisms of Res in SH-SY5Y cells. (**a**) Western blot analysis of NGB levels in SH-SY5Y cells pretreated with (R,R)-THC (10^−6^ M) and Endoxifen (Endox, 10^~6^ M) inhibitors, 1 h before administration, for 24 h, of either vehicle (DMSO 1:1000, *v*:*v*), E2 (10^−9^ M), DPN (10^−9^ M), PPT (10^−9^ M), and the co-treatment with E2 + (R,R)-THC and E2 + Endox at the same concentrations. Tubulin protein was used to normalize blot loading. The upper panel shows a Western blot type, and the lower panel shows the densitometric analysis of at least three different experiments. Data are reported as mean ± SD. *p* < 0.05 was calculated with Student’s *t*-test with respect to the vehicle (*) or with respect to E2 (°). (**b**) Western blot analysis of NGB levels in SH-SY5Y cells pretreated as described in panel a. Tubulin protein was used to normalize blot loading. The upper panel shows a Western blot type, and the lower panel shows the densitometric analysis of at least three different experiments. Data are reported as mean ± SD. *p* < 0.01 was calculated with Student’s *t*-test with respect to vehicle (*), E2 (°), and Res (+). (**c**) Western blot analysis of NGB levels in SH-SY5Y cells treated for 24 h with vehicle (DMSO 1:1000, *v*:*v*) or with E2 (10^~9^ M) or with Res (10^−7^ M) or co-treated with E2 + Res at the same concentrations. Tubulin protein is used to normalize blot loading. The upper panel shows a Western blot type, and the lower panel shows the densitometric analysis of at least three different experiments. *p* < 0.01 was calculated with Student’s *t*-test with respect to vehicle (°) or with respect to E2 (*).

**Figure 3 ijms-24-05903-f003:**
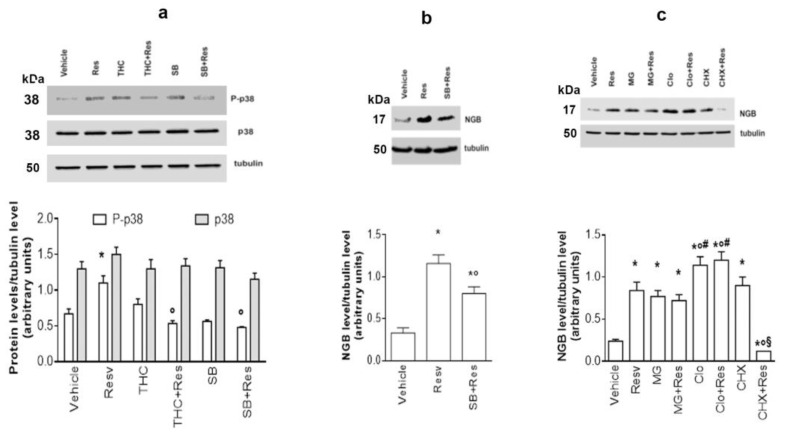
Signaling pathways involved in Res effect on NGB in SH-SY5Y cells. (**a**) Western blot analysis of phosphorylated-p38 (P-p38) and p38 levels in SH-SY5Y cells pretreated with (R,R)-THC (10^−6^ M) and SB (5 × 10^−6^ M) inhibitors 1 h before administration, for 24 h, of vehicle (DMSO 1:1000, *v*:*v*), Res (10^−7^ M) and of the co-treatment with Res + (R,R)-THC and Res + SB at the same concentrations. Tubulin protein was used to normalize blot loading. The upper panel shows a Western blot type, and the lower panel shows the densitometric analysis of six different experiments. Data are reported as mean ± SD. *p* < 0.01 was calculated with Student’s *t*-test with respect to the vehicle (*) or with respect to Res (°). (**b**) Western blot analysis of NGB levels in SH-SY5Y cells treated with vehicle (DMSO 1:1000, *v*:*v*) or Res (10^−7^ M) in the absence or the presence of the p38 inhibitor SB (5 × 10^~6^ M), 1 h pretreatment. Tubulin protein was used to normalize blot loading. The upper panel shows a Western blot type, and the lower panel shows the densitometric analysis of at least three different experiments. Data are reported as mean ± SD. *p* < 0.01 was calculated with Student’s *t*-test with respect to the vehicle (*) and Res (°). (**c**) Western blot analysis of NGB levels in SH-SY5Y cells treated for 4 h with vehicle (DMSO 1:1000, *v*:*v*), Res (10^−7^ M), MG (10^−6^ M), Clo (10^−5^ M) and CHX (10^−5^ M) or treated with Res + MG, Res + Clo, and Res + CHX at the same concentrations. Tubulin protein was used to normalize blot loading. The upper panel shows a Western blot type, and the lower panel shows the densitometric analysis of at least three different experiments. Data are reported as mean ± SD. *p* < 0.01 was calculated with Student’s *t*-test with respect to vehicle (*), Res (°), MG (#), or CHX (§).

**Figure 4 ijms-24-05903-f004:**
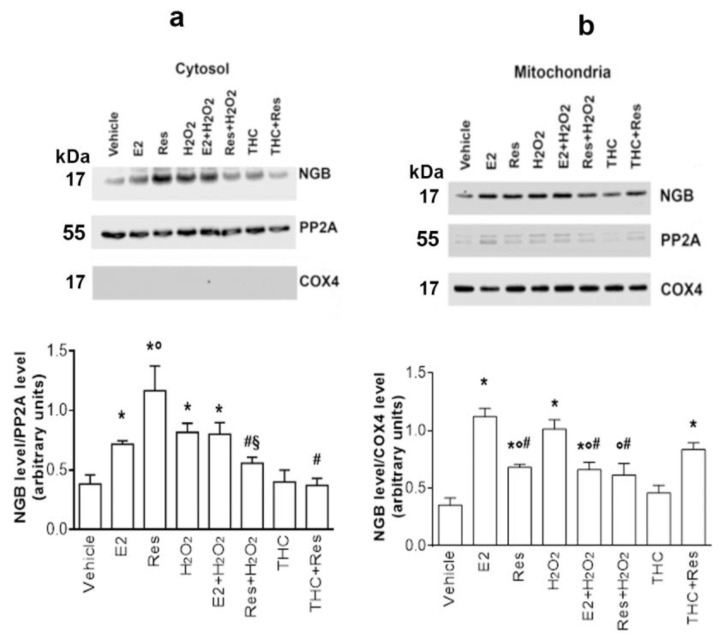
Effect of Res on NGB localization. (**a**) Western blot analysis of NGB levels in SH-SY5Y cytosolic portion, treated with vehicle (DMSO 1:1000, *v*:*v*), E2 (10^−9^ M), Res (10^−7^ M), H_2_O_2_ (5 × 10^−5^ M), and E2 + H_2_O_2_, Res + H_2_O_2_ and (R,R)-THC + Res treatments at the same concentrations; (R,R)-THC (10^−6^ M) was administrated 1 h before the other treatments (24 h). PP2A and COX4 protein levels were used to confirm the purity of the isolated compartments (cytosol and mitochondria, respectively). The upper panel shows a Western blot type, and the lower panel shows the densitometric analysis of at least three different experiments. Data are reported as mean ± SD. *p* < 0.01 was calculated with Student’s *t*-test with respect to vehicle (*), E2 (°) or Res (#), or H_2_O_2_ (§). (**b**) Western blot analysis of NGB levels in the mitochondrial fraction in SH-SY5Y cells, treated as reported in panel a. The upper panel shows a Western blot type, and the lower panel shows the densitometric analysis of at least three different experiments. Data are reported as mean ± SD. *p* < 0.01 was calculated with Student’s *t*-test with respect to vehicle (*), E2 (°), and H_2_O_2_ (#).

**Figure 5 ijms-24-05903-f005:**
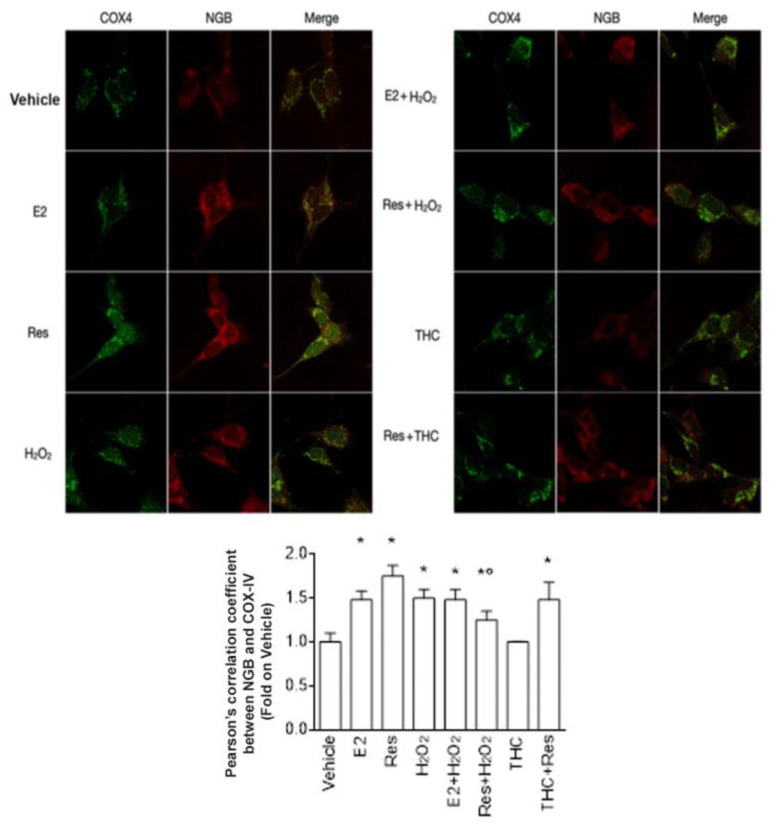
Effect of Res on NGB mitochondrial localization. (Upper panels) Confocal microscopy analysis of NGB/COX4 colocalization in SH-SY5Y cells treated with vehicle (DMSO 1:1000), E2 (10^−9^ M), Res (10^−7^ M), H_2_O_2_ (5 × 10^−5^ µM), and E2 + H_2_O_2_, Res + H_2_O_2_ and (R,R)-THC + Res treatments at the same concentrations; (R,R)-THC (10^−6^ M) was administrated 1 h before the other treatments (24 h). NGB was labeled in red, while COX4 was in green. (Bottom panel) shows the Pearson’s correlation coefficient, calculated with respect to the vehicle-treated samples, of at least six different experiments. Data are reported as mean ± SD. *p* < 0.01 was calculated with Student’s *t*-test with respect to vehicle (*) or Res (°).

**Figure 6 ijms-24-05903-f006:**
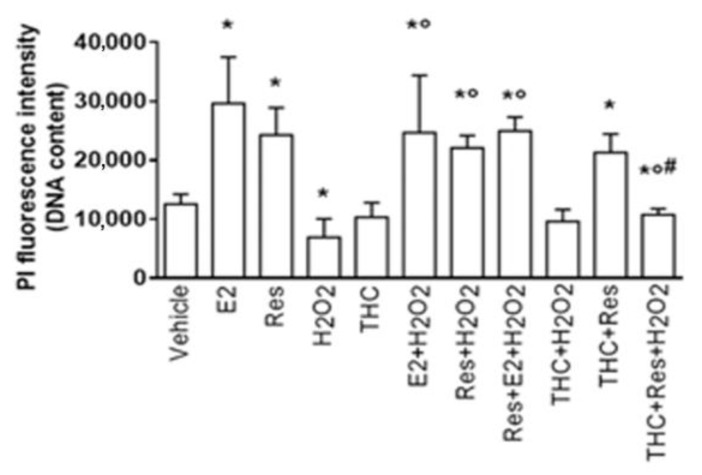
E2 and Res protective effect on SH-SY5Y cell vitality against H_2_O_2_. SH-SY5Y cells were treated with (R,R)-THC (10^−6^ M) for 1 h before 24 h stimulation with vehicle (DMSO 1: 1000, *v*:*v*), E2 (10^−9^ M), Res (10^−7^ M) and H_2_O_2_ (5 × 10^−5^ M). Cell vitality has been evaluated by the propidium iodide (PI) assay, as described in the Section 4. The data represent the mean ± SD of at least six different experiments. *p* < 0.01 was calculated with Student’s *t*-test with respect to the cells treated with vehicle (*), H_2_O_2_ (°), and Res (#).

**Figure 7 ijms-24-05903-f007:**
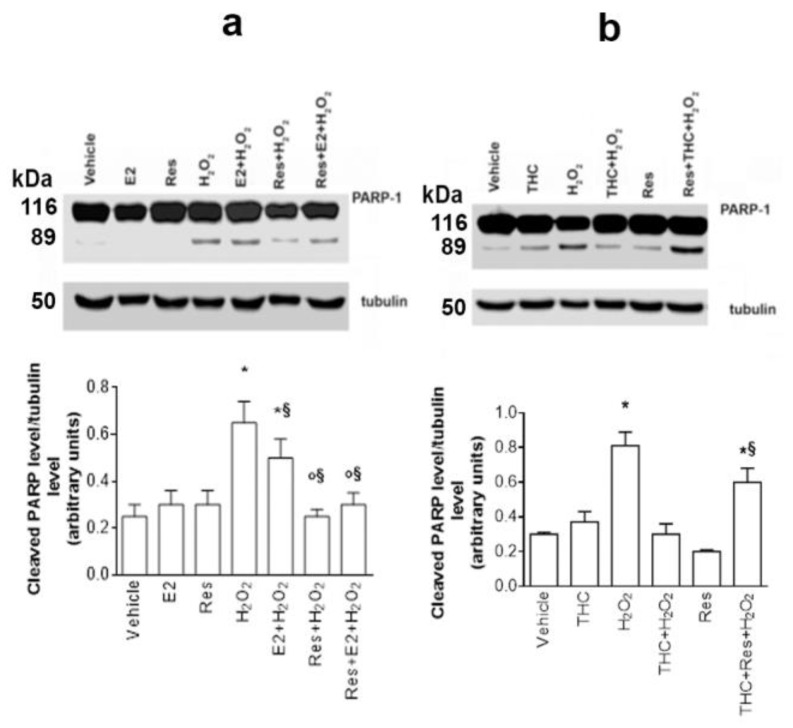
Effect of Res on H_2_O_2_-induced apoptosis in SH-SY5Y cells. (**a**) Analysis by Western blot of PARP-1 cleavage in SH-SY5Y cells treated for 24 h with vehicle (DMSO 1: 1000, *v*:*v*), E2 (10^−9^ M), Res (10^−7^ M), H_2_O_2_ (5 × 10^−5^ M), E2 + H_2_O_2,_ and Res + H_2_O_2_, at the same concentrations. Tubulin protein was used to normalize blot loading. The top panel shows a Western blot type, and the bottom panel shows the densitometric analysis of at least three different experiments. Data are reported as mean ± SD. *p* < 0.01 was calculated with Student’s *t*-test with respect to vehicle (*), E2 (°), and H_2_O_2_ (§). (**b**) Analysis of PARP-1 cleavage in SH-SY5Y cells pretreated with (R,R)-THC (10^−6^ M) 1 h before stimulation with vehicle (DMSO 1: 1000, *v*:*v*), H_2_O_2_ (5 × 10^−5^ M), Res (10^−7^ M), (R,R)-THC + H_2_O_2_, Res + THC, and Res + THC + H_2_O_2_ at the same concentrations for 24 h. Tubulin protein was used to normalize blot loading. The top panel shows the Western blot type, and the bottom panel shows the densitometric analysis of at least three different experiments. Data are reported as mean ± SD. *p* < 0.01 was calculated with Student’s *t*-test with respect to vehicle (*) or Res (§).

**Figure 8 ijms-24-05903-f008:**
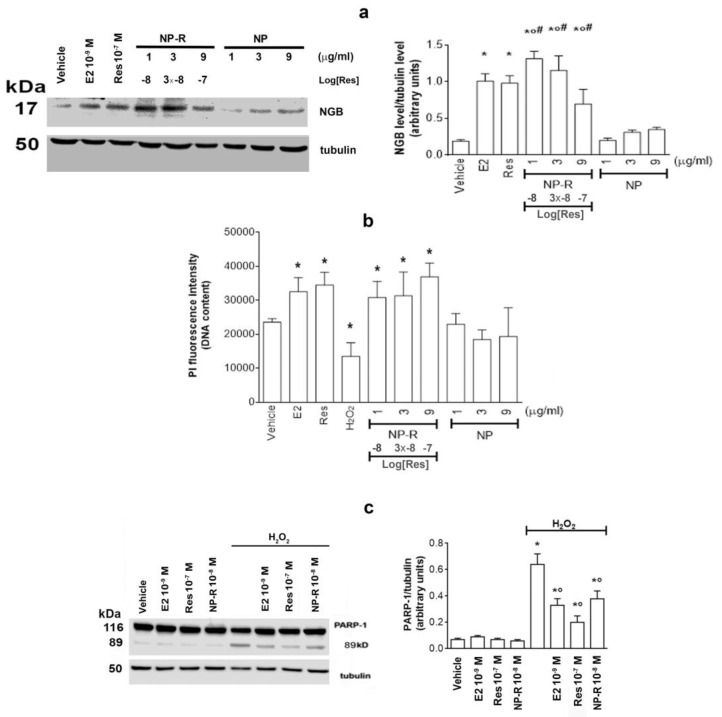
Effects of NP-R in SH-SY5Y Cells. (**a**) Analysis by Western blot of NGB levels in SH-SY5Y cells treated for 24 h with vehicle (DMSO 1: 1000, *v*:*v*), E2 (10^−9^ M), Res (10^−7^ M), NP-R (1 to 9 μg/mL of NP corresponding to a load of Res concentration, respectively, of 10^−8^, 3 × 10^~8^, and 10^−7^ M) and different concentrations of unconjugated NP (1 to 9 μg/mL). The left panel shows a Western blot type, and the right panel shows the densitometric analysis of at least three different experiments. Tubulin protein was used to normalize blot loading. Data are reported as mean ± SD. *p* < 0.05 was calculated with Student’s *t*-test with respect to the vehicle (*) and Res (°) and unconjugated NP (#). (**b**) PI assay of NP-R in SH-SY5Y. Cells were treated as described in panel a. The data represent the mean ± SD of at least four different experiments. *p* < 0.05 was calculated with Student’s *t*-test with respect to the cells treated with vehicle (*). (**c**) Analysis by Western blot of PARP-1 levels in SH-SY5Y cells treated with vehicle (DMSO, 1:1000, *v*:*v*), E2 (10^−9^ M), RSV (10^−7^ M), NP-R (1 to 9 μg/mL, Res loaded concentration of 10^−8^ M), and H_2_O_2_ (5 × 10^−5^ M). The left panel shows a Western blot type, and the right panel shows the densitometric analysis of at least three different experiments. Tubulin protein was used to normalize blot loading. Data are reported as mean ± SD. *p* < 0.05 was calculated with Student’s *t*-test with respect to vehicle (*) and H_2_O_2_ (°).

## Data Availability

The data presented in this study are available within the article.

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
