# Peer review of "A Novel Resveratrol-Induced Pathway Increases Neuron-Derived Cell Resilience against Oxidative Stress"

_ijms, 2023, doi:10.3390/ijms24065903_

Round 1
Reviewer 1 Report
In this manuscript, authors show evidence about the effect of resveratrol on neuron-derived cell resilience against oxidative stress induced by H2O2. In my opinion this study needs some experiments to clearly demonstrate the effect of RES in these cells and the dependence on the activation of ER-beta.
Major points.
1.- Authors indicate along the text that resveratrol is more efficient than E2 in increasing NGB levels in SH-SY5Y cells. However, the concentration of Resveratrol used in these experiments is 100 times higher than E2, thus, the effect of Res cannot be more efficient than E2 since its effect needs 100 times more concentration than the natural ligand of ER.
2.- Authors show that the effect of Res on the expression of NGB is through the activation of ERbeta because THC pretreatment inhibits the effect induced by Res and also the phosphorylation of p38. However, THC can directly inhibit phosphorylation of p38 induced by other ligands such as NMDA (https://www.sciencedirect.com/science/article/pii/S0304394005008359). In fact, in figure 3, authors does not show the results of the combination of E2 with THC in order to determine if the effect of E2 is by activating p-p38 through ER-dependent mechanisms.
3.- In order to determine if the effect of Res depends on the binding and activation of ERbeta experiments performed in ERbeta depleted or KO cells must be performed. If not, the results found with THC pretreatment are inconclusive.
4.- Experiments performed with p38 inhibitor only indicate that p38 is the key component involved in NGB expression induction but not that the effect depends on ER activation by Res. It is known that Res induces p38 phosphorylation in other systems (https://pubmed.ncbi.nlm.nih.gov/11245472/; https://www.nature.com/articles/aps2014132). For this reason, ER-KO experiments are essential to determine the dependence of ER for the Res effect.
5.- Further, in the original blots shown by the authors, p-p38 signal is deficient. The blot shows high levels of non-specific signal and the signal for p-p38 is so low that only a clear revision of the image changing light and contrast can permit the image shown by authors in figure 3. This result is not clear.
6.- CHX effect is not clear. How is that inhibition of protein synthesis is not affected in non-treated cells? This means that the effect of an inhibitor of protein synthesis is affecting more Res treated cells than normal cells. Is it because Res is decreasing the synthesis. The other possibility is the induction of the elimination of the protein but the experiments with MG or Clo do not indicate that. Checking the original image the blots can be bad assigned.
7.- The difference in the effect of Res and E2 in NGB translocation to mitochondria is not chear and the reason why the effect of resveratrol is different than E2 is not correctly explained. If NGB expression is induced by Res, why it does not translocate to mitochondria?
8.- Authors perform confocal experiments and try to quantify the amount of NGB in mitochondria but confocal experiments are not accurate for quantifying proteins in organella, they are only indicated for localization.
9.- In cell viability experiments, I suggest to use MTT or other analysis such as TUNEL or Annexin V/Pi for determination of cell viability. Res is known to delay cell cycle progression and the cells used in this experiment are tumoral cells that can suffer the effect of Res.
10.- Authors use THC combination with Res to explain the ER-dependent effect on the protection against oxidative stress induced by H2O2. However, authors do not take into consideration that THC can decrease GPx levels as has been demonstrated in other cells (Cantarini et al. Int J. Mol Sci, 2020 21(15):5533). As Catalase and GPx protects cells agains H2O2, the depletion of these enzymes can explain why THC inhibits the protective effect of Res against H2O2 injury. Further, it is also known that Res induces GPx expression and activity. For this reason, levels of the enzymes SOD, Catalase and GPx must be determined.
11.- Another puzzling result is that authors indicate that Res and E2 produces a protective effect through inducing and translocating NGB to mitochondria but H2O2 also produces the same effect (figure 4) but the cell vitality experiments demonstrate a damage (figure 6). The relationship between NGB and protection is not clear.
12.- The experiments performed with gold particles does not make sense. Authors indicate that they were performed in order to increase bioavailability of Res but in cell cultures it is not necessary and cannot be extrapolated to experiments with organisms. Further, why NGB translocation to mitochondria was not performed in these experiments?
13.- Discussion is too large and confusing. The first two paragraphs can be reduced. Again authors indicate that Res is more effective than E2 although with 100 times more concentration and without inducing NGB translocation to the mitochondria (lines 402 to 405). They indicate that THC is a selective inhibitor of ERb but also is an inhibitor of p38 phosphorylation by other mechanisms and produce reduction of Gpx, effects that strongly affect these experiments without the intervention of ER.
Minor points.
1.- Indicate MW in Western blotting images.
2.- In image 3c, the original blot shows one blot more than the image shown in figure 3c.
Author Response
In this manuscript, authors show evidence about the effect of resveratrol on neuron-derived cell resilience against oxidative stress induced by H2O2. In my opinion this study needs some experiments to clearly demonstrate the effect of RES in these cells and the dependence on the activation of ER-beta.
Major points.
1.- Authors indicate along the text that resveratrol is more efficient than E2 in increasing NGB levels in SH-SY5Y cells. However, the concentration of Resveratrol used in these experiments is 100 times higher than E2, thus, the effect of Res cannot be more efficient than E2 since its effect needs 100 times more concentration than the natural ligand of ER.
REPLY: We agree with this referee’s comment. Therefore, although Figure 1 shows that 10-7 M Res is more efficient than E2 10-7 M in increasing NGB levels, the sentence has been amended in all text.
2.- Authors show that the effect of Res on the expression of NGB is through the activation of ERbeta because THC pretreatment inhibits the effect induced by Res and also the phosphorylation of p38. However, THC can directly inhibit phosphorylation of p38 induced by other ligands such as NMDA (https://www.sciencedirect.com/science/article/pii/S0304394005008359). In fact, in figure 3, authors does not show the results of the combination of E2 with THC in order to determine if the effect of E2 is by activating p-p38 through ER-dependent mechanisms.
REPLY: As reported in the references 6 and 23 cited in the text, we already demonstrated that E2 via ERβ activates p38 phosphorylation. Here, we would assess if Res triggered the same pathway as E2 do. In order to demonstrate the involvement of ERβ we utilized a selective ERβ inhibitor: (R,R)-5,11-Diethyl-5,6,11,12-tetrahydro-2,8-chrysenediol (THC, Ki = 3.6 nM) (see the Tocris Bioscience site at https://www.tocris.com/search?keywords=THC and references cited in the product datasheet). The compound reported in the referee’s citation is Δ9-tetrahydrocannabinol which is a partial agonist for cannabinoid receptors nor, as far as we know, a modulator of estrogen receptors. The mistake originates from the similar acronym assigned to these two different molecules (i.e., THC). To avoid any confusion, we added in the text the full name of ERβ inhibitor the first time it is cited and changed the acronym in the text to (R,R)-THC.
3.- In order to determine if the effect of Res depends on the binding and activation of ERbeta experiments performed in ERbeta depleted or KO cells must be performed. If not, the results found with THC pretreatment are inconclusive.
REPLY: As mentioned before, the molecule we used to block ERβ activities was misunderstood. In this paper, we used a specific and selective ERβ inhibitor (i.e., (R,R)-THC). In other words, we preferred the pharmacological approach (i.e., (R,R)-THC treatment) instead of the genetic approach (i.e., ER-KO) to evaluate the ability of resveratrol to mimic the E2-induced pathway that protects neurons from apoptosis being both well-known, comparable and widely used approaches to study the involvement of molecules in cell functions.
4.- Experiments performed with p38 inhibitor only indicate that p38 is the key component involved in NGB expression induction but not that the effect depends on ER activation by Res. It is known that Res induces p38 phosphorylation in other systems (https://pubmed.ncbi.nlm.nih.gov/11245472/; https://www.nature.com/articles/aps2014132). For this reason, ER-KO experiments are essential to determine the dependence of ER for the Res effect.
REPLY: We agree to the referee comment: the experiments with the p38 inhibitor clearly indicate that p38 activation is one of the key components involved in Resveratrol-induced NGB accumulation (Figure 3b). In addition, figure 3a clearly demonstrated that the cell pre-treatment with the selective ERβ inhibitor (i.e., (R,R)-THC) completely prevents p38 activation demonstrating that at concentration of 0.1 µM, in neuronal-derived cells, resveratrol triggers via ERβ the activation of p38 important for NGB up-regulation and to prevent apoptosis induced by oxidative stress. The two papers cited by the referee indicate that resveratrol concentration 100 times higher (i.e., 20 or 60 µM) than that used in our paper (i.e., 0.1 µM) could activate a pro-apoptotic signal that involves p38 activation in lymphoblast or epidermal cells. Even if it is possible that different resveratrol concentrations could drive cells to different destiny as already demonstrated (doi: 10.1016/j.freeradbiomed.2022.01.010), our result clearly indicates that resveratrol does not induce cell number decrease nor apoptosis in neuronal cells (see figures 6, 7a, 7b, 8b, and 8c). Therefore, the signals suggested by the referee as activated in lymphoblast or epidermal cells that drive cells to the apoptosis are not working in neurons stimulated with low resveratrol concentrations, which, instead, trigger the anti-apoptotic ERβ/p38/NGB pathway.
5.- Further, in the original blots shown by the authors, p-p38 signal is deficient. The blot shows high levels of non-specific signal and the signal for p-p38 is so low that only a clear revision of the image changing light and contrast can permit the image shown by authors in figure 3. This result is not clear.
REPLY: As per other anti-phospho antibodies, the level of non-specific bands is high also for anti-phospho-p38 antibody. However, we repeat the experiments reported in figure 3a loading the Western Blotting Protein Standards (BioRad) in a lane close to the samples. As reported in the new original blot, although the non-specific bands are still present, now the image is clear and the position of P-p38 is clearly defined by the position of standard band (i.e., 37 kDa). The data of these new experiments have been added to the previous and the statistical analysis have been performed confirming our previous result. We hope that these novel experiments are more credible.
6.- CHX effect is not clear. How is that inhibition of protein synthesis is not affected in non-treated cells? This means that the effect of an inhibitor of protein synthesis is affecting more Res treated cells than normal cells. Is it because Res is decreasing the synthesis. The other possibility is the induction of the elimination of the protein but the experiments with MG or Clo do not indicate that. Checking the original image the blots can be bad assigned.
REPLY: We would thank the referee for this comment that allows us to better clarify these experiments. Although the main aim of this work was to verify the ability of low concentration of resveratrol to mimic the E2-induced ERβ-dependent pathway, a synthesis of this answer has been added in the discussion section. The human Ngb gene codifies for an 1885 bps unique transcript, which cDNA sequence is highly conservative. The transcription start site (TSS) of human Ngb is located at −306 bp from its translation start codon ATG. The Ngb promoter lacks the TATA box; however, many conserved putative transcription factor binding sites have been identified within the human Ngb promoter, including two GC-boxes and two neuron-restrictive silencer element (NRSE) sites (i.e., NRSE1 and NRSE2). These NRSE sites are bound by the neuron-restrictive silencer factor (NRSF), a 210 kDa glycoprotein containing nine zinc finger domains, which can also act as a silencer of neuron-specific gene expression in undifferentiated neuronal progenitor cells. In addition, the Ngb promoter region is characterized by the presence of binding sites for several transcription factors like the Specificity protein (Sp) family members Sp1 and Sp3, the cAMP response element binding (CREB) protein, the early growth response protein 1, members of the Nuclear Factor κ-light chain enhancer of activated B cells (NF-κB) family (i.e., p50, p65, cRel) (doi: 10.1016/j.mam.2016.10.004. Epub 2016 Nov 4). These data describe NGB as a compensatory protein, which levels, in neurons, are maintained at low amounts by the neuron-restrictive silencer factor (NRSF) to increase upon stress or hormonal stimuli reach the cell. Our results are in line with this vision. Indeed, 24h of cycloheximide treatment preventing the translation of the silencer protein could enhance NGB levels, in addition, cycloheximide alters protein degradation through activation of protein kinase B (AKT) (doi: 10.1074/jbc.M112.445148) further enhancing NGB levels. On the other hand, resveratrol triggers the p38 activation, a well-known inducer of CREB activation (doi: 10.1126/scisignal.abq5389. Epub 2022 Dec 13; doi: 10.1016/j.jchemneu.2022.102179. Epub 2022 Oct 29; doi: 10.1038/sj.cr.7290257) that could overcome NRSF inhibition allowing Ngb transcription. Although resveratrol stimulation prevents NGB degradation and enhances Ngb transcription, the contemporary treatment with cycloheximide reduces the NGB translation and the NGB protein overexpression could not occur.
Finally, we forgotten to indicate in the original, uncropped, image that the cell lysate after resveratrol and cycloheximide co-treatment was loaded in the last two lines. The original image has been correctly amended.
7.- The difference in the effect of Res and E2 in NGB translocation to mitochondria is not chear and the reason why the effect of resveratrol is different than E2 is not correctly explained. If NGB expression is induced by Res, why it does not translocate to mitochondria?
REPLY: Figures 4b and 5 clearly demonstrate that resveratrol induces a significant NGB accumulation in mitochondria. All text has been checked to better clarify this result. E2 and resveratrol effects on NGB accumulation in mitochondria are different because E2 binds to and activates both ER subtypes (i.e., ERα and ERβ) being ERα the main subtype responsible for E2-induced NGB translocation in mitochondria (see refs. 6 and 8). On the other hand, although resveratrol also binds both ER subtypes, the polyphenol is an antagonist of ERα inhibiting its activities, and an agonist of ERβ (see ref. 22).
8.- Authors perform confocal experiments and try to quantify the amount of NGB in mitochondria but confocal experiments are not accurate for quantifying proteins in organella, they are only indicated for localization.
REPLY: The amount of NGB in cells after resveratrol stimulation has been studied with the fractionation kit (Figures 4a and 4b), which enables the separation of cytosol and mitochondria compartments. PP2A and COX4 protein levels were used to confirm the purity of the isolated compartments (cytosol and mitochondria, respectively). NGB level in mitochondria was also confirmed with confocal microscopy, evaluating the co-localization of NGB (red) with the mitochondrial protein COX4 (green) (Figure 5). The use of ImageJ software was used to compare the results obtained with two different assays (i.e., cell fractionation and confocal microscopy).
9.- In cell viability experiments, I suggest to use MTT or other analysis such as TUNEL or Annexin V/Pi for determination of cell viability. Res is known to delay cell cycle progression and the cells used in this experiment are tumoral cells that can suffer the effect of Res.
REPLY: In our work, we used the Propidium Iodide assay to determine cell viability along with the PARP-1 cleavage to monitor, specifically, the activation of intrinsic apoptosis. In particular, PI is a fluorescent intercalating agent that can be used to stain cells and DNA by intercalating between the bases allowing the measure of total cell content. PARP-1 is cleaved specifically by caspase3 so that it is no longer active, preventing DNA repair and promoting apoptotic processes. The assay suggested by the referee do not add any new information to that obtained from the assays used in our work. Indeed, the MTT assay is used to measure the metabolic activity of cells, which is an indicator of cell proliferation, cell viability, and cytotoxicity. TUNEL assay is a method for detecting DNA fragmentation and thus apoptosis. The Annexin V/PI protocol is a commonly used approach for studying apoptotic cells. Each assay has its pros and cons even those suggested by the referee; thus, to avoid possible artifacts in this work we performed two assays to state that resveratrol increases the vitality of neurons and protects them from oxidative stress.
10.- Authors use THC combination with Res to explain the ER-dependent effect on the protection against oxidative stress induced by H2O2. However, authors do not take into consideration that THC can decrease GPx levels as has been demonstrated in other cells (Cantarini et al. Int J. Mol Sci, 2020 21(15):5533). As Catalase and GPx protects cells agains H2O2, the depletion of these enzymes can explain why THC inhibits the protective effect of Res against H2O2 injury. Further, it is also known that Res induces GPx expression and activity. For this reason, levels of the enzymes SOD, Catalase and GPx must be determined.
REPLY: As already mentioned, here we used the (R,R)-THC not the Δ9-tetrahydrocannabinol as reported in the paper of Cerretani et al. Int J. Mol Sci, 2020 21(15):5533 that we believe has been cited by the referee. The search of (R,R)-5,11-Diethyl-5,6,11,12-tetrahydro-2,8-chrysenediol AND oxidative stress did not produce any results in PubMed, while in Google Scholar 36 paper were found, in which this compound has been used as ERβ inhibitor to demonstrate the involvement of this receptor subtype in estradiol (E2)-induced protection against oxidative stress (doi: 10.1016/j.jsbmb.2010.12.004). It could be possible that the effect of resveratrol on GPx is mediated via ERβ, but this is very far from the aim of our study.
11.- Another puzzling result is that authors indicate that Res and E2 produces a protective effect through inducing and translocating NGB to mitochondria but H2O2 also produces the same effect (figure 4) but the cell vitality experiments demonstrate a damage (figure 6). The relationship between NGB and protection is not clear.
REPLY: Neuroglobin is a protein that acts as a stress sensor. Different kinds of stressors including nutrient deprivation, hypoxia, and oxidative stress enhance NGB levels. In particular high ROS levels (i.e., H2O2) increase NGB amounts in mitochondria and, in parallel, the cytochrome c released from cardiolipin. In this context, NGB action is impaired and the cell fails to achieve protection. E2, resveratrol, as well as other NGB inducers, increase the amount of globin before the stress, thus when ROS levels increase, NGB is already accumulated in the mitochondria and could bind to cytochrome c reducing its release from mitochondria and allowing cell resilience to the stress.
12.- The experiments performed with gold particles does not make sense. Authors indicate that they were performed in order to increase bioavailability of Res but in cell cultures it is not necessary and cannot be extrapolated to experiments with organisms. Further, why NGB translocation to mitochondria was not performed in these experiments?
REPLY: The sense of the experiments with gold nanoparticles was to determine if conjugated resveratrol, which has a good solubility and is less prone to the gut and liver metabolism, had a similar effect than unconjugated compound in inducing NGB accumulation and protection against oxidative stress. Our data indicate that a low concentration of resveratrol (0.01 µM) is required to accumulate NGB and to protect cells from apoptosis. Thus, it is possible to further reduce the resveratrol concentration to obtain an effect. The huge amount of literature demonstrating that only in the mitochondria NGB could maintain its anti-apoptotic role, prompted us do not looking for NGB localization after the same stimulus (i.e., resveratrol). The sense of these experiments is to check the toxicity of gold nanoparticles and to find the most effective concentration of conjugated resveratrol, data that cannot be obtained in animals, at least in our country, where the 3 Rs principle (i.e., Replacement, Reduction and Refinement) is an ethical requirement for scientists.
13.- Discussion is too large and confusing. The first two paragraphs can be reduced. Again authors indicate that Res is more effective than E2 although with 100 times more concentration and without inducing NGB translocation to the mitochondria (lines 402 to 405). They indicate that THC is a selective inhibitor of ERb but also is an inhibitor of p38 phosphorylation by other mechanisms and produce reduction of Gpx, effects that strongly affect these experiments without the intervention of ER.
REPLY: We reduced the discussion trying to clarify the concept expressed. However, the sentence of resveratrol efficacy with respect to E2 has been amended, Resveratrol induces an accumulation of NGB in mitochondria, and we used (R,R)-THC as ERβ inhibitor, not Δ9-tetrahydrocannabinol in our experiments.
Minor points.
1.- Indicate MW in Western blotting images.
REPLY: The MW have been inserted
2.- In image 3c, the original blot shows one blot more than the image shown in figure 3c.
REPLY: The original blot of figure 3c has been correctly reported
Reviewer 2 Report
The paper entitled “A novel Resveratrol-induced pathway increases neuron-derived 4 cell resilience against oxidative stress” describes the effect of resveratrol through the estrogen receptor beta (ER β). The authors used a very well-designed experimental design to show the pathway activated by resveratrol in SH-SY5Y cells; the results obtained are sound, clear, and convincing. However, I think that more than a Western blot should be used to demonstrate some of the conclusions.
1. Fig. 3. Using the proteasome inhibitor (MG-132), the lysosomal degradation inhibitor chloroquine, and the inhibitor of gene translation (cycloheximide), this experiment also showed the alteration on NGB and p-p38 levels after treatment using Western blot and demonstrated of the mechanism of action. These are no complete experiments since reference 24 showed that E2 activates AKT and PKC to inhibit the degradation of NGB. No experiments associated with these enzymes are shown. Additionally, to clearly show that resveratrol prevents degradation additional experiments are needed. The experiment showed that both MG-132 and chloroquine have the same effect, which is some imprecise. Additionally, Cycloheximide has a 100% effect, then, how resveratrol could inhibit degradation if the increased levels are affected by the new synthesis? I believe that an extensive analysis of the mechanism is needed.
gene translation the measured of the mRNA NGB levels is required.
2. Figs. 4 and 5. Effect of NGB localization. Western blot was used to demonstrate the effect of resveratrol on NGB localization at cytosol or mitochondria (cells were). Although the effect is evident, I am not sure if a Western blot it’s enough to demonstrate that a different amount of protein inside the mitochondria has an important effect. This experiment was complemented with immunofluorescence. I suggest that the images should show a greater number of cells to clearly shows the effect.
3. Fig. 6. Represents the effect of resveratrol in vitality against H2O2 using propidium iodide assay. The graph should be accompanied by a photograph that shows the nuclei of the cells, to evaluate the structure of nuclei (condense, entire, pyknotic, etc.). Probably, can be complemented with other forms of evaluated apoptosis addition to PARP-1 (shown in figure 7).
4. Fig. 8 showed the effect of resveratrol conjugated with nanospheres in protection against oxidative stress. This experiment tries to show that the conjugated form of resveratrol has an increased bioavailability, however, this property should be evaluated using vascular micro endothelium in a model of the blood-brain barrier. I don’t understand why the conjugated form has an increased effect on cultures. It suppose that the binding to the receptor was increased.
Minor:
Add the number of catalogs of the antibodies used in the study.
Author Response
The paper entitled “A novel Resveratrol-induced pathway increases neuron-derived 4 cell resilience against oxidative stress” describes the effect of resveratrol through the estrogen receptor beta (ER β). The authors used a very well-designed experimental design to show the pathway activated by resveratrol in SH-SY5Y cells; the results obtained are sound, clear, and convincing. However, I think that more than a Western blot should be used to demonstrate some of the conclusions.
- Fig. 3. Using the proteasome inhibitor (MG-132), the lysosomal degradation inhibitor chloroquine, and the inhibitor of gene translation (cycloheximide), this experiment also showed the alteration on NGB and p-p38 levels after treatment using Western blot and demonstrated of the mechanism of action. These are no complete experiments since reference 24 showed that E2 activates AKT and PKC to inhibit the degradation of NGB. No experiments associated with these enzymes are shown. Additionally, to clearly show that resveratrol prevents degradation additional experiments are needed. The experiment showed that both MG-132 and chloroquine have the same effect, which is some imprecise. Additionally, Cycloheximide has a 100% effect, then, how resveratrol could inhibit degradation if the increased levels are affected by the new synthesis? I believe that an extensive analysis of the mechanism is needed. gene translation the measured of the mRNA NGB levels is required.
REPLY: We would thank the referee for this comment that allows us to better clarify these experiments. Although the main aim of this work was to verify the ability of low concentration of resveratrol to mimic an E2-induced ERβ-dependent pathway, a synthesis of this answer has been added in the discussion section. Reference 24 is related to the effect of E2 in breast cancer cells, which express only the ERα subtype. In neuron-derived cells (reference 23) we clearly demonstrated that E2-induced NGB accumulation is exclusively dependent on ERβ-mediated p38 activation. To avoid repeating already published data, here the involvement of AKT and PKC, which activation is mediated by ERα, has not been performed. However, reference 23 has been added to the sentence cited by the referee. MG-132 and chloroquine are two well-known inhibitors of proteasomal and lysosomal, respectively, protein degradation. The reported data indicate that, like E2, resveratrol-induced NGB degradation depends on both proteasomal and lysosomal pathways. Indeed, pre-treating cells with proteasomal inhibitor, MG-132, an increase in NGB level occurs that resveratrol cannot prevent or further increase due to the lysosomal degradation is still working and vice versa. We are sorry, but it is not clear to us the meaning of ‘imprecise’ related to this experiment.
Finally, the result obtained by treating cells with cycloheximide. As reported in the reply to the comment 6 of the 1 reviewer, the human Ngb gene codifies for an 1885 bps unique transcript, which cDNA sequence is highly conservative. The transcription start site (TSS) of human Ngb is located at −306 bp from its translation start codon ATG. The Ngb promoter lacks the TATA box; however, many conserved putative transcription factor binding sites have been identified within the human Ngb promoter, including two GC-boxes and two neuron-restrictive silencer element (NRSE) sites (i.e., NRSE1 and NRSE2). These NRSE sites are bound by the neuron-restrictive silencer factor (NRSF), a 210 kDa glycoprotein containing nine zinc finger domains, which can also act as a silencer of neuron-specific gene expression in undifferentiated neuronal progenitor cells. In addition, the Ngb promoter region is characterized by the presence of binding sites for several transcription factors like the Specificity protein (Sp) family members Sp1 and Sp3, the cAMP response element binding (CREB) protein, the early growth response protein 1, members of the Nuclear Factor κ-light chain enhancer of activated B cells (NF-κB) family (i.e., p50, p65, cRel) (doi: 10.1016/j.mam.2016.10.004. Epub 2016 Nov 4). These data describe NGB as a compensatory protein, which levels, in neurons, are maintained at low amounts by the neuron-restrictive silencer factor (NRSF) to increase upon a stress or hormonal stimuli reach the cell. Our results are in line with this vision. Indeed, 24h of cycloheximide treatment preventing the translation of the silencer protein could enhance NGB levels, in addition, cycloheximide alters protein degradation through activation of protein kinase B (AKT) (doi: 10.1074/jbc.M112.445148) further enhancing NGB levels. On the other hands, resveratrol triggers the p38 activation, a well-known inducers of CREB activation (doi: 10.1126/scisignal.abq5389. Epub 2022 Dec 13; doi: 10.1016/j.jchemneu.2022.102179. Epub 2022 Oct 29; doi: 10.1038/sj.cr.7290257) that could overcome NRSF inhibition allowing Ngb transcription. Although resveratrol stimulation prevents NGB degradation and enhances Ngb transcription, the contemporary treatment with cycloheximide reduces the NGB translation and the NGB protein overexpression could not occur.
- Figs. 4 and 5. Effect of NGB localization. Western blot was used to demonstrate the effect of resveratrol on NGB localization at cytosol or mitochondria (cells were). Although the effect is evident, I am not sure if a Western blot it’s enough to demonstrate that a different amount of protein inside the mitochondria has an important effect. This experiment was complemented with immunofluorescence. I suggest that the images should show a greater number of cells to clearly shows the effect.
REPLY: This is not a single experiment, but 3 experiments in double on cell fractionation and 6 experiments in double on confocal microscopy have been performed, thus a great number of cells have been analyzed. Both approaches give the same results as reported in the densitometries strongly sustaining our statement.
- Fig. 6. Represents the effect of resveratrol in vitality against H2O2 using propidium iodide assay. The graph should be accompanied by a photograph that shows the nuclei of the cells, to evaluate the structure of nuclei (condense, entire, pyknotic, etc.). Probably, can be complemented with other forms of evaluated apoptosis addition to PARP-1 (shown in figure 7).
REPLY: In our work, we used the Propidium Iodide assay to determine cell viability along with the PARP-1 cleavage to monitor specifically the activation of intrinsic apoptosis. In particular, PI is a fluorescent intercalating agent that can be used to stain cells and DNA by intercalating between the bases. PARP-1 is cleaved specifically by caspase3 so that it is no longer active, preventing DNA repair and promoting apoptotic processes. The other assay suggested by the referee could not add any new information to that obtained from the assays used in our work.
- Fig. 8 showed the effect of resveratrol conjugated with nanospheres in protection against oxidative stress. This experiment tries to show that the conjugated form of resveratrol has an increased bioavailability, however, this property should be evaluated using vascular micro endothelium in a model of the blood-brain barrier. I don’t understand why the conjugated form has an increased effect on cultures. It suppose that the binding to the receptor was increased.
REPLY: A specific strategy is needed to preserve resveratrol from the extensive metabolism to which polyphenols are exposed, and to maintain its protective effects. A common way to increase the bioavailability of drugs is to improve the aqueous solubility of compounds through their functionalization. Several kinds of RSV functionalization have been reported in the literature and a variety of chemically different capping groups and pro-moieties have been used to modulate the absorption and release of this molecule, as well as to improve its biological activity (doi:10.3390/antiox8080244). In this study, we would determine if resveratrol conjugated with gold nanoparticles, which has a good solubility and is less prone to the gut and liver metabolism as suggested by literature, had a similar effect than unconjugated compound in inducing NGB accumulation and protection against oxidative stress. Moreover, the goal of these experiments was to check the cell toxicity of gold nanoparticles and to find the most effective concentration of conjugated resveratrol, data that cannot be obtained in animals, at least in our country.
We recently reported the ultrastructural analysis by Focused Ion Beam/Scanning Electron Microscopy (FIB/SEM) of breast cancer cells treated with nanopartcles(DOI: 10.3390/ijms24032148), in, that. Treatment with vehicle was used as control. A large number of nano-particles were localized to the cytosol after 24 h treatment in samples treated with NP or NP-R. Both naked and conjugated nanoparticles displayed the tendency to aggregate with some of the largest macrocomplexes located inside vacuoles for their possible elimination. Noteworthy, cells exposed to nanoparticles exhibited similar ultrastructural features to control cells. In fact, in all the analyzed conditions cells showed a regular plasma membrane with microvilli, a rounded nucleus with a smooth or slightly irregular nuclear envelope, and abundant mitochondria displaying an even distribution through-out the cytoplasm and regular cristae arrangement. The remaining organelles (e.g., endoplasmic reticulum and Golgi apparatus) showed normal ultrastructural features and cytoplasmic distribution irrespective of the different conditions investigated. As a whole these data suggest that an great level of resveratrol could be discharged from nanoparticles with respect to unconjugated resveratrol that should go through plasma membrane to reach the cytoplasm-localized receptor.
Minor:
Add the number of catalogs of the antibodies used in the study.
REPLY: the number of catalog of antibodies has been added.
Round 2
Reviewer 1 Report
I have read carefully the comments of the authors and consider that the manuscript needs revision in deep. In general, authors have only modified the text and some images of the Western blotting but not more than these small modifications.
For example, in the comment 11, authors only respond with an explanation about the role of NGB in mitochondria but they do not explain why the behavior of this protein is different after E2 and Res induction. Most of the indications of the authors are referred to other works but they do not check them. Experiments with KO cells for ERalfa or ER beta are needed in order to determine if the effect of Res is ER-dependent or not. As I indicated in my previous comments, Res can induce phosphorylation of p38 by ER-independent mechanisms and this can explain the effect of Res in these cells without the intervention of ER.
The pharmacological inhibition of ER by THC (now explained by the authors) is not enough to determine that the effect of Res is especific of its interaction with ERbeta. Res has been shown to bind and activate many different and non-related enzymes and proteins. The specificity of the activation of this pathway is important in order to reduce the high number of signaling pathways involved in its physiological effect.
This is why I consider many of my indications have not been correctly explained.
Author Response
I have read carefully the comments of the authors and consider that the manuscript needs revision in deep. In general, authors have only modified the text and some images of the Western blotting but not more than these small modifications. REPLY: We sincerely do not understand the reason of this referee’s comment. Many referee’s comments were sustained from its mistake to consider the specific ERβ inhibitor we used (i.e., R,R-THC, as reported in all manuscript) as Δ9-tetrahydrocannabinol (the principal psychoactive constituent of cannabis that binds to endocannabinoid receptors located in the cerebral cortex, cerebellum, and basal ganglia), which has no relation to estrogen receptors. In particular, the referee’s comments 2, 3, 4, 10, 13 and the literature cited by the referee is related to the effects of Δ9-tetrahydrocannabinol that has no relation with our work. Since the referee thought we had used tetrahydrocannabinol and not a specific ERbeta inhibitor, he rightly asked us to KO the receptor. In our reply, we clarified the misunderstanding, also modifying the text, and commented that since we had used a specific inhibitor, which is widely used in the specific literature, the ERbeta silencing data would not add any new information to what we had already demonstrated. To render more clearly our experiments also the text has been amended accordingly. The referee's comment 1 pointed out a possible misunderstanding about comparing different concentrations of E2 and resveratrol and this has been corrected in the text. The referee’s comment 5 pointed to a low-quality of fig. 3a. Therefore, we repeated the western blot experiments and inserted the new experiments as requested. Thus, new experiments have been performed not just a modification of an image, that has been performed in the original, uncropped western blot. The referee’s comment 6 requests a clarification of the results obtained with cycloheximide treatment. We explained better these experiments both in the answer to the referee and in the text considering that what results no enough clear to the referee could be also unclear for the readers. Comment 7 required another clarification on the differences in NGB translocation to the mitochondria induced by Res and E2. In this comment there is another referee’s misunderstand (reported also in comment 13) in reading our work. Indeed, he asked why resveratrol does not induce the translocation of NGB into the mitochondrion. In our answer, we explained the difference in the responses between E2 and resveratrol, already present in the text, thus the text was not amended, and referred the referee to figures 4b and 5, which clearly demonstrate the ability of resveratrol to translocate NGB into the mitochondrion. In comments 8 and 9 the referee suggest the use of other analyses (MTT, TUNEL or Annexin V) to determine cell viability instead of PI assay and PARP-1 cleavage that we used. As far as we know, there are no compulsory assays to assess the presence of a phenomenon, in our case cell viability, several assays can be used from MTT to XTT to cell counting. Each laboratory uses the assay that has been developed in the years. In this specific case, because there are several of them, we have used two different assays, well known to the entire scientific community. The other assays suggested by the referee could not add any new information to that obtained from the assays used in our work. In the referee’s comment 11, he asked to clarify the relationship between NGB and neural protection. This clarification has been added in the answer. Finally, in the comment 12 the referee stated that the experiments with nanoparticles made no sense. In the response, which is extensively present in the text, the logic of the experiments was explained. Overall, of the 13 comments made by the referee, 5 arose from a misunderstanding which we hope will be definitively clarified, 5 requested clarifications which were punctually included in the text and 3 requested new experiments. The one requested in comment 5 has been performed and added in the manuscript, the other requested principally in the comment 9, in our opinion are redundant and require more founds and working time that not all laboratory can afford. On the base of this analysis, we do not understand the referee’s comment ‘[Authors] have only modified the text and some images of the Western blotting but not more than these small modifications’.
For example, in the comment 11, authors only respond with an explanation about the role of NGB in mitochondria but they do not explain why the behavior of this protein is different after E2 and Res induction. REPLY: This explanation is reported in the reply to referee’s comment 7 in which the explanation has been required as well as in the text.
Most of the indications of the authors are referred to other works but they do not check them. REPLY: Most of our indications become from our previous data that we, as other scientists, do not need to repeat and repeat again. We used these previous observations to make a hypothesis that has been verified with novel experiments. The Scientific Method do not requires that previous observations should be confirmed especially when derive from the same lab. As example, we already demonstrated, in 2010, by using both silencing and inhibitor of ERbeta, that E2-induced NGB accumulation in neuron-derived cells requires ERbeta/p38 activation. Thus, here we used the ERbeta and p38 inhibitors to verify our hypothesis that also resveratrol could utilize the same pathway.
Experiments with KO cells for ERalfa or ER beta are needed in order to determine if the effect of Res is ER-dependent or not. REPLY: the ERalpha silencing is a novelty of these comments; unfortunately, the referee does not add any justification to its comment, thus it is difficult to discuss any reply. Moreover, as reported in the reply to the previous referee’s comment 3, the molecule we used to block ERβ activities was misunderstood. In this paper, we used a specific and selective ERβ inhibitor (i.e., (R,R)-THC). In other words, we preferred the pharmacological approach (i.e., (R,R)-THC treatment) instead of the genetic approach (i.e., ER-KO) to evaluate the ability of resveratrol to mimic the E2-induced pathway that protects neurons from apoptosis being both well-known, comparable and widely used approaches to study the involvement of molecules in cell functions. We do not consider a mistake or inaccuracy or imprecision to choose one approach instead of another, on the contrary we prefer the pharmacological approach because of we are sure that 100% of ERbeta is blocked, while the success of ERbeta silencing in neuronal-derived cells is around 70%, as already demonstrated by us. Moreover, the pharmacological approach maintain better the physiology of the cells that could be strongly modified by the transient transfection of genetic materials. If the referee knows any statement of any scientific community that disagree or dissuade from the use of pharmacological approach, we promptly will change our belief.
As I indicated in my previous comments, Res can induce phosphorylation of p38 by ER-independent mechanisms and this can explain the effect of Res in these cells without the intervention of ER. REPLY: the publications reported by the referee in the comment 4, in which this comment was stated, are related to other resveratrol concentration 100 times higher (i.e., 20 or 60 µM) than that used in our paper (i.e., 0.1 µM). These citations indicate that resveratrol-induced p38 activation could activate a pro-apoptotic signal that involves p38 activation in lymphoblast or epidermal cells. Even if it is possible that different resveratrol concentrations could drive cells to different destiny as already demonstrated (doi: 10.1016/j.freeradbiomed.2022.01.010), our result clearly indicates that resveratrol does not induce cell number decrease nor apoptosis in neuronal cells (see figures 6, 7a, 7b, 8b, and 8c). Therefore, the signals suggested by the referee as activated in lymphoblast or epidermal cells that drive cells to the apoptosis are not working in neurons stimulated with low resveratrol concentrations, which, instead, trigger the anti-apoptotic ERβ/p38/NGB pathway.
The pharmacological inhibition of ER by THC (now explained by the authors) is not enough to determine that the effect of Res is especific of its interaction with ERbeta. Res has been shown to bind and activate many different and non-related enzymes and proteins. The specificity of the activation of this pathway is important in order to reduce the high number of signaling pathways involved in its physiological effect. This is why I consider many of my indications have not been correctly explained. REPLY: Authors have explained in the first version of the manuscript that they used the pharmacological inhibition of ERbeta (it is specific); in the revised version, we changed the acronym to avoid confusion in the reader. What does mean ‘pharmacological inhibition is not enough’? What does mean ‘is important … to reduce the high number of signaling pathways involved in [resveratrol] effect’? The resveratrol ability to bind to ERs has been already demonstrated more than 20 years ago (doi: 10.1210/endo.141.10.7721.) even if the downstream signal pathways have not yet described. In our work, we identify a downstream pathway that is very common for ERbeta (i.e., p38 activation) and culminate with NGB accumulation into mitochondria protecting neuronal derived cells from apoptosis. It is clear that this referee does not believe in our data and that he would conduct the experiments with other approaches. In any case we feel that the requests made by the referee are not justified by the controls we carried out for every piece of data obtained. We still consider that our conclusions on the effect of low, nutritionally significant concentrations of resveratrol in neuronal-derived cells, which drive cells to the resistance against oxidative stress-induced apoptosis, are sustained by the data obtained. These results could change the vision of the resveratrol effects and could open new avenues to bring this promising molecule to the clinical trials, but the referee seems not interested in this aspect.
Reviewer 2 Report
The authors added the required information and responded to my comments and doubts in detail. I have no further comments.
Author Response
We would thank the referee for his/her comments that helped us to ameliorate our manuscript.
Round 3
Reviewer 1 Report
This is the third revision of this manuscript and the main issues about the problems of this study remains because authors have not introduced essential studies to demostrate the fact that the effect of resveratrol on these cells is ERb dependent. I recognize my first mistake confounding THC with Δ9-tetrahydrocannabinol instead of R,R-THC in part because the authors did not described the compound correctly in the first version of their article and in part because tetrahydrocannabinol is also an inhibitor or ERb as authors must know (Suso Takeda, Biol Pharm Bull, 2014;37(9):1435-8. doi: 10.1248/bpb.b14-00226).
1.- In this new version authors insist in the higher efficiency of resveratrol in comparison with E2 (lines 89 and 90) but this is not true. In figure 12, the concentrations used of E2 and Resveratrol are shown in different WBs and makes difficult to compare the efficiency in the increase of NGB. Further, in figure 1c the concentrarion of E2 used is not indicated and in this figure the efficiency is clearly not so higher in the case of Resveratrol. But, this is clearer in figure 2c in which authors already indicate the concentrations of E2 (10-9) and Resveratrol (10-7) used. Quantification of this effect clearly demonstrate that with 100 times less concentration, the effect of E2 is only half of resveratrol. This indicates that E2 is more efficient than resveratrol. Further, in figure 4a, 10-9 M E2 shows twice increase in comparison with control whereas 10-7 M resveratrol only increases to around 3 times levels of NGB in comparison with control. Clearly, these figures demonstrate that the effect of E2 is stronger than resveratrol since with 100 times less concentration the effect is only 1/2 of resveratrol or even more. Authors wants to insist that resveratrol is more efficient than E2 but the results do not demonstrate this fact.
2.- Quantification of WBs must be carefully cheched. For example, in Figure 2b, the signal of NGB in E2 and Res treatment is clearly in comparison with control. However, the signal of Resv + endotoxin is even higher than E2 signal accordingly to the quantification but in the image the blot is clearly lower. Authors indicate that this means that ERa is not affected by resveratrol but the blot indicate that is clearly affected accordingly with the signal found with Res alone. Further, why is possible that the signal of Res + THC is the same than control or THC alone while the blot clearly indicate that this signal is higher than vehicle, endoc and THC+Endox? In summary, quantification not always fits with the blots. The same problems can be found in figure 4a in which the signal with H2O2 is clearly higher than the signal with E2 but the quantification shows similar levels of even E2+H2O2. Figure 4b also shows similar problems. Please, compare E2 with H2O2. With lower COX4 levels and similar NGB signal, the induction of NGB in mitochondria is the same in both columns.
3.- In the case of p-p38 authors does not provide the ration p-p38/p38 that is the only way to demonstrate the activation of a protein by phosphorylation. THC p-p30 signal is clearly increased in comparison with vehicle, however, the quantification, again shoes a si ilar level than control that shows clearly a very low level of phosphorylation. Intringly, when combined with resv, the levels of phosphorylation decrease. If both, Resv and THC increase the phosphorylation of p-38, why the levels decrease when combined. Probably the role of THC as inducer of ERa is involved in this aspect. In the same figure, the effect of CHX, that inhibits protein synthesis in combination of Resv is very confuse because CHX did not inhibit the levels of this protein, even increase them in comparison with vehicle. This does not make sense.
4.- As I indicated in previous comments, confocal images cannot be used to quantify the levels of proteins, they can be used for colocalization but as quantification of fluorescence depends on the time of exposition to laser or UV light and can decay with time, the quantification between different samples cannot be used, so, quantification in figure 5 does not make sense.
5.- Authors try to demonstrate that resveratrol protects cells agains cytotoxicity induced by H2O2 measuring PI signal in 96 well microplate. This experiment cannot be used to quantify cells. In fact, the results indicate that in only 24 hours E2 and resveratrol were able to accelerate the growth of cells two or three times. This is a very high increase in growth but resveratrol is known to reduce cell growth in these cells at the concentration used (DOI:10.1159/000328516). I consider tha better quantification method even the direct counting of cells must be used instead of PI fluorescence. Further, if we compare the results of figure 7 in which E2+H2O2 does not protect against apoptosis with the results in figure 6 in which E2+H2O2 does not show differenes with E2 alone and a clear protective effect is shown, the discrepancy between the results with PI and with PARP-1 is clear. Further, in figure 7b authors show that THC+H2O2 show reduced levels of PARP-1 break but in figure 6, the growth of cells is clearly affected and not protection vs H2O2 effect is found, again a clear discrepancy between the results. Why THC+Resv results are not shown in figutre 7b?
6.- In figure 8a, again the clear similar effect of E2 vs Resveratrol showing the same levels of NGB induction with 100 times less concentration of E2. Clearly, the indication that Resv is more efficient of E2 is not based on the results of this paper.
7.- Interestingly (R,R)-THC is an antagonist of ERb but, at the same time an agonist of ERa. Authors indicate in discusion (lines 399-403) that activation of ERa can inhibit the activation of the p38 pathway through the activation of the PI3K/AKT and ERK/MAPK signaling pathways. Thus, the effect found in p-p38 with resveratrol in presence of THC can be associated with the activation of ERa by this compound instead of inhibition of ERb. It is known that Resveratrol induces p-p38 by other mechanisms (Cancer Res. 2001 Feb 15;61(4):1604-10) and probaby the interaction of THC with ERa and the activation of other pathways reduce this phosphorylation explaining all the effect of resveratrol with independence of ERb. For this reason I insist, the only way to demonstrate the ERb-dependent effect of resveratrol is through KO ERb.
Author Response
This is the third revision of this manuscript and the main issues about the problems of this study remains because authors have not introduced essential studies to demostrate the fact that the effect of resveratrol on these cells is ERb dependent. I recognize my first mistake confounding THC with Δ9-tetrahydrocannabinol instead of R,R-THC in part because the authors did not described the compound correctly in the first version of their article and in part because tetrahydrocannabinol is also an inhibitor or ERb as authors must know (Suso Takeda, Biol Pharm Bull, 2014;37(9):1435-8. doi: 10.1248/bpb.b14-00226).
REPLY: As reported in the first reply to this referee (Reply to question 2) we do not find any paper in PubMed reporting the effect of Δ9-tetrahydrocannabinol as an inhibitor of ERbeta activities. The paper cited by referee stated “Δ9-THC positively stimulates the ERβ expression, which results in the inhibition of E2/ERα signaling” that, in our opinion does not mean that Δ9-THC is an inhibitor of ERbeta activities, but confirms the role of dominant negative of ERbeta levels on ERalpha transcriptional activities.
1.- In this new version authors insist in the higher efficiency of resveratrol in comparison with E2 (lines 89 and 90) but this is not true. In figure 12, the concentrations used of E2 and Resveratrol are shown in different WBs and makes difficult to compare the efficiency in the increase of NGB. Further, in figure 1c the concentrarion of E2 used is not indicated and in this figure the efficiency is clearly not so higher in the case of Resveratrol. But, this is clearer in figure 2c in which authors already indicate the concentrations of E2 (10-9) and Resveratrol (10-7) used. Quantification of this effect clearly demonstrate that with 100 times less concentration, the effect of E2 is only half of resveratrol. This indicates that E2 is more efficient than resveratrol. Further, in figure 4a, 10-9 M E2 shows twice increase in comparison with control whereas 10-7 M resveratrol only increases to around 3 times levels of NGB in comparison with control. Clearly, these figures demonstrate that the effect of E2 is stronger than resveratrol since with 100 times less concentration the effect is only 1/2 of resveratrol or even more. Authors wants to insist that resveratrol is more efficient than E2 but the results do not demonstrate this fact.
REPLY: The authors have chosen not to emphasize the effectiveness of resveratrol in relation to E2, as this does not alter the meaning of our findings. Consequently, we have revised all relevant sections of the text accordingly.
2.- Quantification of WBs must be carefully cheched. For example, in Figure 2b, the signal of NGB in E2 and Res treatment is clearly in comparison with control. However, the signal of Resv + endotoxin is even higher than E2 signal accordingly to the quantification but in the image the blot is clearly lower. Authors indicate that this means that ERa is not affected by resveratrol but the blot indicate that is clearly affected accordingly with the signal found with Res alone. Further, why is possible that the signal of Res + THC is the same than control or THC alone while the blot clearly indicate that this signal is higher than vehicle, endoc and THC+Endox? In summary, quantification not always fits with the blots. The same problems can be found in figure 4a in which the signal with H2O2 is clearly higher than the signal with E2 but the quantification shows similar levels of even E2+H2O2. Figure 4b also shows similar problems. Please, compare E2 with H2O2. With lower COX4 levels and similar NGB signal, the induction of NGB in mitochondria is the same in both columns.
REPLY: We analyzed again all our western blots repeating the densitometric analyses. The data for each individual western blot were obtained by calculating the ratio of the band intensity (in pixels) of the protein of interest (e.g., NGB, p38) to the band intensity (in pixels) of the housekeeping protein (e.g., tubulin, PP2A), which served as a normalization control for loading variations. We then used Student's t-test to determine the statistical significance between several blots comparing two different samples. The statistical significance between samples is reported in the figure legend. It should be noted that we compared not only the treatment with the control, but also different treatments with each other. Finally, we selected the most representative western blot image after completing our data analysis. Therefore, it is statistically unlikely that the densitometric measurements shown in the figures are identical to the corresponding lanes, as even a small increase in the intensity of the tubulin band could result in a decrease in the densitometric analysis of the protein of interest.
3.- In the case of p-p38 authors does not provide the ration p-p38/p38 that is the only way to demonstrate the activation of a protein by phosphorylation. THC p-p30 signal is clearly increased in comparison with vehicle, however, the quantification, again shoes a si ilar level than control that shows clearly a very low level of phosphorylation. Intringly, when combined with resv, the levels of phosphorylation decrease. If both, Resv and THC increase the phosphorylation of p-38, why the levels decrease when combined. Probably the role of THC as inducer of ERa is involved in this aspect. In the same figure, the effect of CHX, that inhibits protein synthesis in combination of Resv is very confuse because CHX did not inhibit the levels of this protein, even increase them in comparison with vehicle. This does not make sense.
REPLY: As the referee knows very well, the way to demonstrate the activation of a signaling protein is to show its phosphorylation status. The data could be reported as a ratio or as we have done being the level of total p-38 always the same in all samples. However, we regret to inform the referee that we did not observe any significant increase in p-38 phosphorylation due to THC treatment. Therefore, it is challenging to discuss non-existent data. Regarding the CHX effect, we provided a detailed explanation in response to the first question of the second referee and the sixth question of this referee in our initial response. It is useless to repeat again the same discussion that seems to be neglected by the referee while has been accepted by the second referee.
4.- As I indicated in previous comments, confocal images cannot be used to quantify the levels of proteins, they can be used for colocalization but as quantification of fluorescence depends on the time of exposition to laser or UV light and can decay with time, the quantification between different samples cannot be used, so, quantification in figure 5 does not make sense.
REPLY: We appreciate the reviewer giving us the opportunity to clarify this point once again. As we explained in our initial response, we utilized confocal microscopy analysis to evaluate the co-localization of NGB (red) with the mitochondrial protein COX4 (green). The graph displayed in Figure 5 is not a representation of protein signal quantification. Rather, it is a quantification of the co-localization between NGB and the mitochondrial marker COX-4. We calculated the Pearson correlation coefficient per cell and expressed it as a ratio of the Vehicle treated sample. Although Pearson's correlation coefficient is not influenced by signal levels and signal offset (background), we kept the image acquisition settings constant for each experiment. We have modified the legend of Figure 5 and the graph to provide more clarity regarding the results.
5.- Authors try to demonstrate that resveratrol protects cells agains cytotoxicity induced by H2O2 measuring PI signal in 96 well microplate. This experiment cannot be used to quantify cells. In fact, the results indicate that in only 24 hours E2 and resveratrol were able to accelerate the growth of cells two or three times. This is a very high increase in growth but resveratrol is known to reduce cell growth in these cells at the concentration used (DOI:10.1159/000328516). I consider tha better quantification method even the direct counting of cells must be used instead of PI fluorescence. Further, if we compare the results of figure 7 in which E2+H2O2 does not protect against apoptosis with the results in figure 6 in which E2+H2O2 does not show differenes with E2 alone and a clear protective effect is shown, the discrepancy between the results with PI and with PARP-1 is clear. Further, in figure 7b authors show that THC+H2O2 show reduced levels of PARP-1 break but in figure 6, the growth of cells is clearly affected and not protection vs H2O2 effect is found, again a clear discrepancy between the results. Why THC+Resv results are not shown in figutre 7b?
REPLY: It is very difficult for researcher working since many years on cell growth to accept the sentence ‘resveratrol is known to reduce cell growth in these cells at the concentration used’ on the basis of the literature cited by referee. Indeed, nowhere in the cited paper, it is reported the effect of Res on SHSY5Y cell growth, but in that paper, which is focused on Res effects on autophagy, it is reported that Res is not pro-apoptotic and that 50 µM Res decreases the cell death induced by rotenone. Moreover, in fig.6 THC alone does not change the cell number (or DNA content) with respect to the vehicle, H2O2 treatment decreases the cell number, and THC+H2O2 restore the cell number to the control level, showing the low protective effect of THC on H2O2-induced cell death, which is more evident in figure 7. We do not discuss this effect because of the aim of our work was to evaluate if Res effects rely on ERbeta and not to characterize THC effects. Regarding the E2 effects, both in Fig. 6 and Fig. 7 the protective effects of E2 against cell death induced by H2O2 are reported, again the referee still refers to the single band of PARP without any consideration on the tubulin band on which the dentitometric analyses are based on.
6.- In figure 8a, again the clear similar effect of E2 vs Resveratrol showing the same levels of NGB induction with 100 times less concentration of E2. Clearly, the indication that Resv is more efficient of E2 is not based on the results of this paper.
REPLY: As told in the reply to question 1 Authors decide to not further insist on the efficiency of resveratrol with respect to E2 and the text has been modified accordingly.
7.- Interestingly (R,R)-THC is an antagonist of ERb but, at the same time an agonist of ERa. Authors indicate in discusion (lines 399-403) that activation of ERa can inhibit the activation of the p38 pathway through the activation of the PI3K/AKT and ERK/MAPK signaling pathways. Thus, the effect found in p-p38 with resveratrol in presence of THC can be associated with the activation of ERa by this compound instead of inhibition of ERb. It is known that Resveratrol induces p-p38 by other mechanisms (Cancer Res. 2001 Feb 15;61(4):1604-10) and probaby the interaction of THC with ERa and the activation of other pathways reduce this phosphorylation explaining all the effect of resveratrol with independence of ERb. For this reason I insist, the only way to demonstrate the ERb-dependent effect of resveratrol is through KO ERb.
REPLY: Our data does not demonstrate any impact of THC alone on p-38 activation. However, we previously reported in Figure 6 that Res functions as an antagonist of ERalpha. This finding suggests that the effect of Res in the presence of THC is exclusively linked to ERbeta. As authors, we are concerned that the referee's comment is influenced by the effects of Res in cancer cells, which are notably distinct from the effects of Res in neuronal-derived cells. We want to reiterate that our data confirms that Res has effects at lower concentrations than those previously employed and that these effects are related to ERbeta activities.